# Regulation of *FT* splicing by an endogenous cue in temperate grasses

Zhengrui Qin[1,*], Jiajie Wu[2,*], Shuaifeng Geng[1], Nan Feng[1], Fengjuan Chen[2], Xingchen Kong[1], Gaoyuan Song[1], Kai Chen[1], Aili Li[1], Long Mao[1] & Liang Wu[1,3]

Appropriate flowering timing is crucial for plant reproductive success. The florigen, FLOWERING LOCUS T (FT), interacts with 14-3-3 proteins and the bZIP transcription factor FD, functioning at core nodes in multiple flowering pathways. There are two FT homologues, FT1 and FT2, in *Brachypodium distachyon*. Here we show that *FT2* undergoes age-dependent alternative splicing (AS), resulting in two splice variants (*FT2α* and *FT2β*). The *FT2β*-encoded protein cannot interact with FD or 14-3-3s but is able to form heterodimers with FT2α and FT1, thereby interfering with the florigen-mediated assembly of the flowering initiation complex. Notably, transgenic plants overproducing FT2β exhibit delayed flowering, while transgenic plants in which *FT2β* is silenced by an artificial microRNA display accelerated flowering, demonstrating a dominant-negative role of FT2β in flowering induction. Furthermore, we show that the AS splicing of *FT2* is conserved in important cereal crops, such as barley and wheat. Collectively, these findings reveal a novel posttranscriptional mode of *FT* regulation in temperate grasses.

[1] National Key Facility for Crop Gene Resources and Genetic Improvement, Institute of Crop Science, Chinese Academy of Agricultural Sciences, Beijing 100081, China. [2] State Key Laboratory of Crop Biology, Shandong Agricultural University, Taian 271018, China. [3] Department of Agronomy, College of Agriculture and Biotechnology, Zhejiang University, Hangzhou 310058, China. * These authors contributed equally to this work. Correspondence and requests for materials should be addressed to L.M. (email: maolong@caas.cn) or to L.W. (email:liangwu@zju.edu.cn).

The correct timing of the transition from the vegetative to the reproductive stage is a critical point during plant life cycles. Plants require appropriate environmental conditions, including optimal photoperiod and temperature, to stimulate flowering at a specific time of the year. Additionally, plants also must develop to a certain age with fixed amounts of hormones to achieve flowering[1]. As the encoder of the mobile flower-promoting signal florigen, the *FLOWERING LOCUS T* (*FT*) gene has been shown to have key roles in integrating external and endogenous cues to control the onset of flowering.

FT proteins are a clade of the phosphatidylethanolamine-binding protein (PEBP) family, which act as highly conserved regulators in plants that relay signals from upstream to downstream in flowering pathways. In *Arabidopsis thaliana*, *FT* messenger RNA (mRNA) gradually increases with developmental time under a long-day (LD) photoperiod[2]. After synthesis in leaves, FT protein travels through the vasculature to the shoot apical meristem (SAM) and interacts with a bZIP transcription factor, named FD, which is expressed specifically in the SAM[3,4]. Subsequently, the FT/FD complex activates a number of MADS box genes, including *APETALA1*, *FRUITFULL* and *SUPPRESSOR OF OVEREXPRESSION OF CONSTANS 1* to stimulate floral organ initiation[5]. The 14-3-3 proteins, a family of regulatory molecules conserved in animal and plants, can directly interact with the FT homologue Heading date 3a in rice apical shoots, forming a functional complex that translocates to the nucleus[6]. Thus, 14-3-3 proteins may function as scaffold proteins to associate FT with FD to trigger flowering.

Numerous *FT* transcriptional regulators have been identified to date. For example, in *A. thaliana*, CONSTANS and GIGANTEA activate *FT* expression in the photoperiod pathway[7–9], while FLOWERING LOCUS C and SHORT VEGETATIVE PHASE form a MADS-box complex that negatively mediates *FT* expression in the flowering vernalization-related pathway[10]. Two well-known plant-conserved microRNAs (miRNAs), miR156 and miR172, directly or indirectly act at the *FT* region through their targets, mediating age-dependent flowering processes[11–13]. Compared with the extensive knowledge of *FT* control at the transcriptional level, our understanding of *FT* regulation at the posttranscriptional level is limited.

Temperate grasses, including wheat and barley, are major sources of food and biofuel worldwide. Some specialized components, lacking orthologues in *A. thaliana* and rice, have been discovered in temperate grasses and can affect *FT* gene expression in day-length and vernalization flowering pathways[14–16]. The typical *Pooideae* grass *Brachypodium distachyon* has been considered as a suitable model system for investigating gene function in temperate cereals because of its small genome, simple growth requirements and short life cycle[17]. Phytochrome C, rather than phytochrome A or B, is an essential light receptor for photoperiodic flowering in wheat and *B. distachyon*, suggesting that there are different evolutionary features of LD flowering responses in temperate grasses[18–20]. Moreover, a *Pooideae*-specific miRNA, miR5200, has recently been identified that directly silences two *FT* orthologues and has a crucial role in the photoperiod-mediated flowering pathway in *B. distachyon*, further implying that a distinct flowering control mechanism exists in temperate cereals[14].

Alternative splicing (AS) is a universal mechanism that produces multiple mRNAs from one gene through the variable selection of splice sites during mRNA generation[21,22]. AS can modulate gene expression levels by introducing a premature termination codon, resulting in the degradation of a special RNA-splicing isoform through a nonsense-mediated decay pathway[23]. Additionally, different transcript isoforms may produce truncated proteins that may have different subcellular localization, stability level or function[23]. Several studies have implicated the involvement of AS in plant development, circadian rhythm, starch metabolism, hormone signaling and abiotic stress response, as well as pathogen defense[24–36], suggesting that this mechanism of gene regulation has a significant role in plant adaptation and evolution. Despite the revelation by next-generation sequencing that AS occurs in >60% of *A. thaliana* and 42% of *B. distachyon* genes, the biological significance of most AS events in plants is still largely unknown[21,37–39].

Previously, we demonstrated that two orthologous *FT* genes, *FTL1* (*Bradi2g07070*) and *FTL2* (*Bradi1g48830*), have miR5200-targeting sites and are regulated by miR5200 in *B. distachyon*[14]. Because in *B. distachyon*, more amino acids of FTL2, compared with FTL1, are identical with those of florigen VRN3 in wheat and barley, we re-designated FTL1 as FT2 and FTL2 as FT1 to be clearer when analyzing the phylogenetic relationships among the FTs of temperate grasses[40].

In this study, we show that AS of *FT2* has an important role in controlling florigen activity by producing a competitive repressor in *B. distachyon*. FT2β, which only lacks a short section of the N-terminal PEBP domain, can attenuate the activities of the flowering initiation complex by interacting with full-size FT2α and FT1. The *FT2β*/*FT2α* ratio progressively decreased with plant growth, resulting in a slow gradual increase in the florigen activity level. The overexpression of *FT2β* in transgenic plants resulted in delayed flowering, whereas plants with downregulated FT2β activity through artificial miRNA-induced repression, exhibited accelerated flowering, further demonstrating that *FT2β* acts as a dominant-negative repressor in flowering time control in *B. distachyon*. Together, these results reveal a novel molecular mode of *FT* posttranscriptional regulation in temperate grasses.

## Results

**FT2 is subject to AS generating two isoforms.** In *B. distachyon*, FT1 and FT2 share high amino-acid sequence identity and their overproduction gives rise to extremely early bolting phenotypes, indicating that they perform redundant roles in flowering initiation[14,15,40]. During the study of the regulatory mechanism of FT1 and FT2 in flowering, we interestingly found that two PCR products were generated by a pair of FT2 primers, implying the existence of *FT2* AS in *B. distachyon* (Fig. 1a). By comparing the two *FT2* cDNA sequences, we discovered that one splicing product was missing 84 nucleotides at the 5′-end of the first exon compared with a typical *FT* gene. Thus we designated this form as *FT2β* and the full-size transcript as *FT2α* (Fig. 1b). A protein sequence alignment showed that the missing 28 amino acids came from the 5′ terminal region of the PEBP domain of FT2β, which is more variable compared with the middle and 3′ regions (Fig. 1c; Supplementary Fig. 1a). However, no *FT1* splicing variant was detected in *B. distachyon*, possibly because the gene structure of *FT1* is different from that of *FT2* (Supplementary Fig. 1b,c).

Using specific primers to amplify each *FT2* RNA splicing product, we detected the presence of both variants in *B. distachyon* in most cases (Fig. 1a). Most AS events in plants are shown to be affected by ambient temperature fluctuations[24,25,28,41]. Similarly, the ratios of *FT2β*/*FT2α* were altered when plants were grown at high temperatures (Supplementary Figs 2 and 3).

To determine the role, if any, of different FT2 variants in the control of flowering, we attempted to independently overproduce FT2α and FT2β in *B. distachyon*. As previously described by us and others, the ectopic expression of intact *FT2* (*FT2α*) in *B. distachyon* leads to very early flowering and arrested vegetative

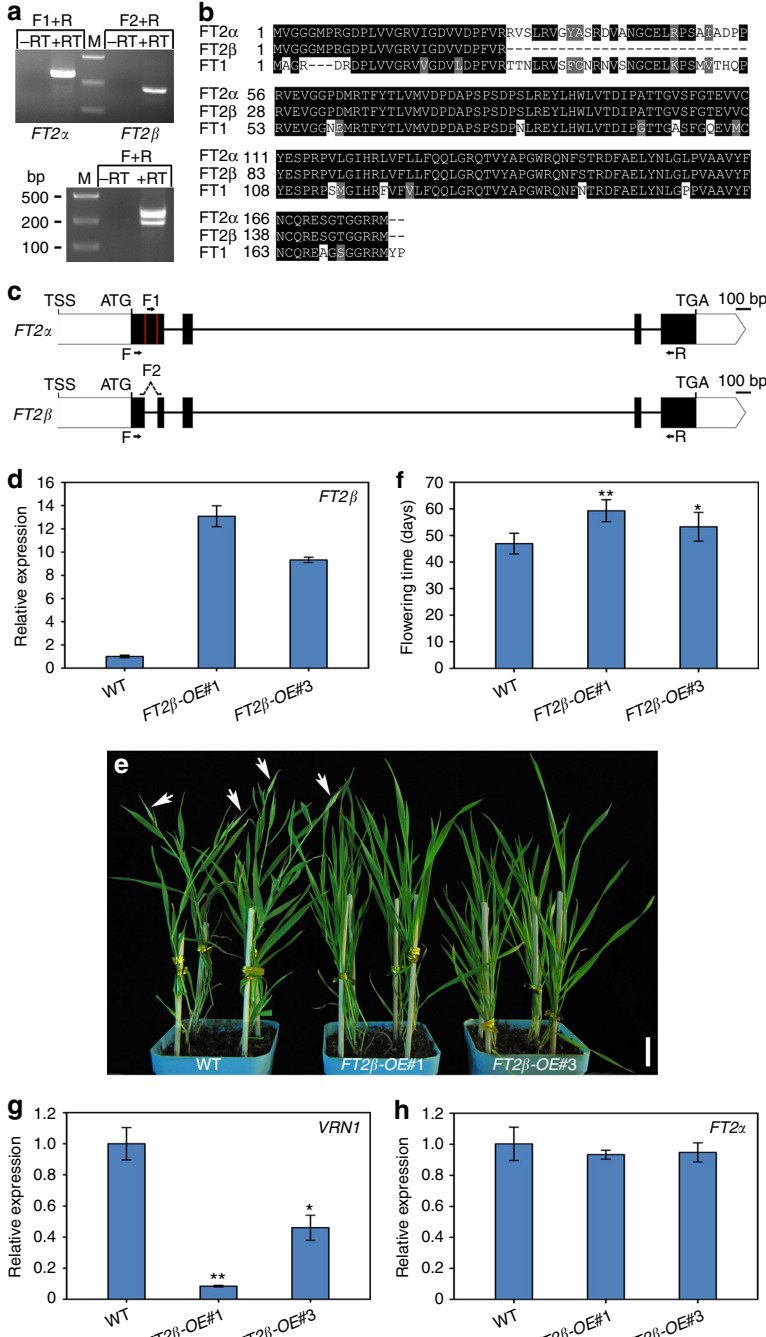

**Figure 1 | Identification of *FT2AS* in *B. distachyon*.** (**a**) Detection of alternatively spliced *FT2* transcripts by RT-PCR. F1 and F2 are specific forward primers for *FT2α* and *FT2β* amplification, respectively, while primer F is used for *FT2α* and *FT2β* PCR simultaneously. R is the reverse primer. RT, reverse transcription. M, marker. bp, base pairs. (**b**) Protein sequence alignment of different FT2 alternative isoforms and FT1. (**c**) Schematic genomic structure of *FT2* splicing variants. White boxes indicate untranslated regions and black boxes indicate exons. TSS indicates the transcriptional start site. (**d**) qRT-PCR analysis of *FT2β* expression in wild-type Bd21 and the indicated *FT2β-OE* transgenic plants. Three four-week-old *B. distachyon* plants were pooled and collected for RNA isolation. *UBC18* was used as an internal control for normalization of qRT-PCR results. qRT-PCR analyses were performed in three biological replicates with similar results. The point represents the mean value of three technical replicates in a representative biological experiment. Error bars indicate s.d. (**e**) Representative phenotype of *FT2β* overexpressing transgenic plants. White arrows point to spikes. Scale bar, 2 cm. (**f**) Flowering time of wild-type and the indicated *FT2β-OE* transgenic plants. Error bars indicate s.d. ($n = 13$). Student's *t*-test, **$P < 0.01$, *$P < 0.05$. (**g**) qRT-PCR analysis of downstream flowering gene *VRN1* expression in wild-type and the indicated *FT2β-OE* transgenic plants. The four-week-old whole plant tissues including leaves and shoot apex were collected for *VRN1* examination. qRT-PCR analyses were performed in three biological replicates with similar results. The point represents the mean value of three technical replicates in a representative biological experiment. Error bars indicate s.d. Student's *t*-test, **$P < 0.01$, *$P < 0.05$. (**h**) qRT-PCR analysis of *FT2α* expression levels in wild-type and the indicated *FT2β-OE* transgenic plants.

growth[14,40], demonstrating the positive florigen activity of FT2α. Next, we generated *FT2β* overexpression (*FT2β-OE*) transgenic plants and determined the flowering time in their $T_1$ generations under LD conditions. Wild-type and transgenic plants displayed similar rates of vegetative development. However, eight independent positive *FT2β-OE* plants exhibited significantly late heading dates compared with wild-type and the null segregants of the transgenic plants (Supplementary Fig. 4).

Because *FT2β-OE#1* and *FT2β-OE#3* displayed a severe and mild delay of flowering time, respectively, we chose their $T_3$ homozygous plants for further molecular and phenotypic analyses. Quantitative real-time PCR (qRT-PCR) showed that *FT2β* was approximately 12- and 9-fold higher in *FT2β-OE#1* and *FT2β-OE#3* plants, respectively, than in the parental wild-type (Fig. 1d; Supplementary Fig. 5a,b). *FT2β-OE#1* bolted 12 days (59 versus 47 days) and *FT2β-OE#3* bolted more than 6 days (53 versus 47 days) later than wild-type plants under LD conditions (Fig. 1e,f). Furthermore, the expression of *VRN1*, an *APETALA1* orthologue acting downstream of FT in temperate grasses, was dramatically reduced in *FT2β-OE#1* and *FT2β-OE#3* plants (Fig. 1g; Supplementary Fig. 5c,d). The *FT2β-OE* plants exhibited wild-type levels of *FT2α* mRNA, excluding the possibility that the late flowering of *FT2β-OE* was caused by the suppression of *FT2α* expression (Fig. 1h; Supplementary Fig. 5e,f). Thus FT2α and FT2β may have different effects on floral transition in *B. distachyon*.

**FT2β does not interact with FD or 14-3-3 proteins**. In *A. thaliana* and rice, during transport from leaves to SAM, florigen interacts with several 14-3-3 proteins in the cytoplasm and subsequently translocates to the nucleus to bind FD and form a flowering activation complex.

The interaction of TaFDL2, the FD-like protein in wheat, with FT1 and FT2 has been previously demonstrated[4,42]. Thus we cloned its orthologue from *B. distachyon* and assigned the same name as in wheat (Supplementary Fig. 6). To address whether FT2α and FT2β in *B. distachyon* can form a flowering initiation complex with FD, we first used yeast two-hybrid assays to determine their interactions with FDL2. Strong interactions were detected between FT2α and FDL2 in yeast cells. However, there was no detectable binding of FT2β to FDL2 in yeast (Fig. 2a). To verify these results, we examined the associations between the two FT2 isoforms and FDL2 using bimolecular fluorescence complementation (BiFC) assays with a split yellow fluorescent protein (YFP) system in *Nicotiana benthamiana* leaf mesophyll cells. As shown in Fig. 2b, there was a strong fluorescent signal in the nucleus when FT2α was transiently co-expressed with FDL2. By contrast, no YFP fluorescence could be detected in leaves co-expressing FT2β and FDL2, suggesting that FT2β cannot bind FDL2 *in vivo*. Co-immunoprecipitation (Co-IP) assays further confirmed a strong binding between FT2α and FDL2 but no interaction between FT2β and FDL2 (Fig. 2c).

Similar to Heading date 3a in rice, wheat FT1 can also interact with two 14-3-3 proteins, GF14b and GF14c[6,42]. Thus we selected several *B. distachyon* 14-3-3 proteins including GF14b and GF14c to determine their abilities to interact with FT2α and FT2β (Fig. 2d; Supplementary Fig. 7a). Both GF14b and GF14c interacted with FT2α in the yeast two-hybrid and BiFC assays (Fig. 2d,e). In contrast, FT2β did not bind to any 14-3-3 proteins (Fig. 2d,e; Supplementary Fig. 7b). Thus 14-3-3 proteins may not form bridges with FT2β to trigger flowering in *B. distachyon*. Together, these observations suggest that FT2 splicing variants may have different actions, with one forming a flowering activation complex as a result of direct physical interactions with 14-3-3s and FD, while the other has lost this ability.

**FT2β attenuates the bindings of florigen to FD and 14-3-3s**. Genetic engineering experiments have shown that unnatural splicing variants can act as dominant-negative mutants to interrupt the activities of wild-type isoforms[24,25,28]. FT2β's inability to bind to FD and 14-3-3s implied that it may interfere with flowering activation complex formation. To test this possibility, we expressed FT2β in yeast cells, which contained BD-FT2α and AD-FDL2 fusions (Fig. 3a). β-Galactosidase (β-Gal) activity assays revealed that the expression of FT2β substantially repressed FT2α-FD binding (Fig. 3b). Likewise, the interaction between FT2α and GF14b in yeast three-hybrid systems was significantly reduced when FT2β was expressed (Fig. 3b).

To further investigate the negative effects of FT2β on flowering activation complex formation, we transiently co-expressed FT2β with split YFP-tagged FT2α and FDL2 proteins in *N. benthamiana* leaf mesophyll cells. Compared with the GUS protein, which was expressed instead of FT2β as a control, the numbers of fluorescent cell nuclei, as well as the intensity of fluorescence, generated by the interaction of FT2α and FDL2 was reduced when FT2β was co-expressed (Fig. 3c,d). Meanwhile, in line with the yeast three-hybrid assay results, competition BiFC experiments also showed a repressive effect of FT2β on the association of FT2α with GF14b in the cytoplasm of *N. benthamiana* leaf cells (Fig. 3e,f).

Because FT1 and FT2 may have redundant roles in flowering initiation[4,14,40], we investigated whether FT2β could affect the efficiency of flowering induction complex formed by FT1, in addition to that formed by FT2α. Indeed, FT2β had an inhibitory effect in the competition BiFC assay (Supplementary Fig. 8). Collectively, these results indicate that FT2β functions as a repressor of the formation of the flowering initiation complex containing FT1, FT2α, 14-3-3s and FD.

**FT2β forms heterodimers with FT2α and FT1**. FT2β suppresses the interactions of florigen proteins with FD and 14-3-3s; however, we did not detect any physical interactions of FT2β with FDL2 or 14-3-3s (Fig. 2). Thus we next investigated another possible mechanism.

It has been suggested that heterodimers may be generated by different splicing isoforms[24,25,28], thus we investigated whether FT2β was able to form a heterodimer with FT2α to produce a nonfunctional protein complex. To test this possibility, we first performed a yeast two-hybrid analysis and observed a strong interaction between FT2α and FT2β (Fig. 4a). The formation of a heterodimer between FT2α and FT2β was also detected in the nuclei as well as in the cytoplasm of *N. benthamiana* leaf cells using BiFC assays (Fig. 4b). Furthermore, a yeast two-hybrid analysis showed an interaction of FT2β with FT1, suggesting that FT2β is able to form heterodimers with both florigen proteins (Supplementary Fig. 9).

Interestingly, yeast two-hybrid and BiFC assays revealed that both FT2α and FT1, but not FT2β, could form homodimers (Fig. 4a–c; Supplementary Fig. 9). Thus we propose that the negative effect of FT2β on flowering time results from the binding of FT2β to FT2α and FT1, which prevents them from interacting with FD and 14-3-3s, thereby reducing the number of functional flowering initiation complexes.

***FT2 AS is controlled by an endogenous cue***. Having discovered a molecular mode by which FT2β represses flowering, we wondered when FT2β contributes to the regulation of floral initiation in *B. distachyon*. Unlike in *A. thaliana*, temperature has a very limited impact on the timing of phase transition from the vegetative to reproductive stage in *B. distachyon*[43]; therefore, *FT2* splicing may have little, if any,

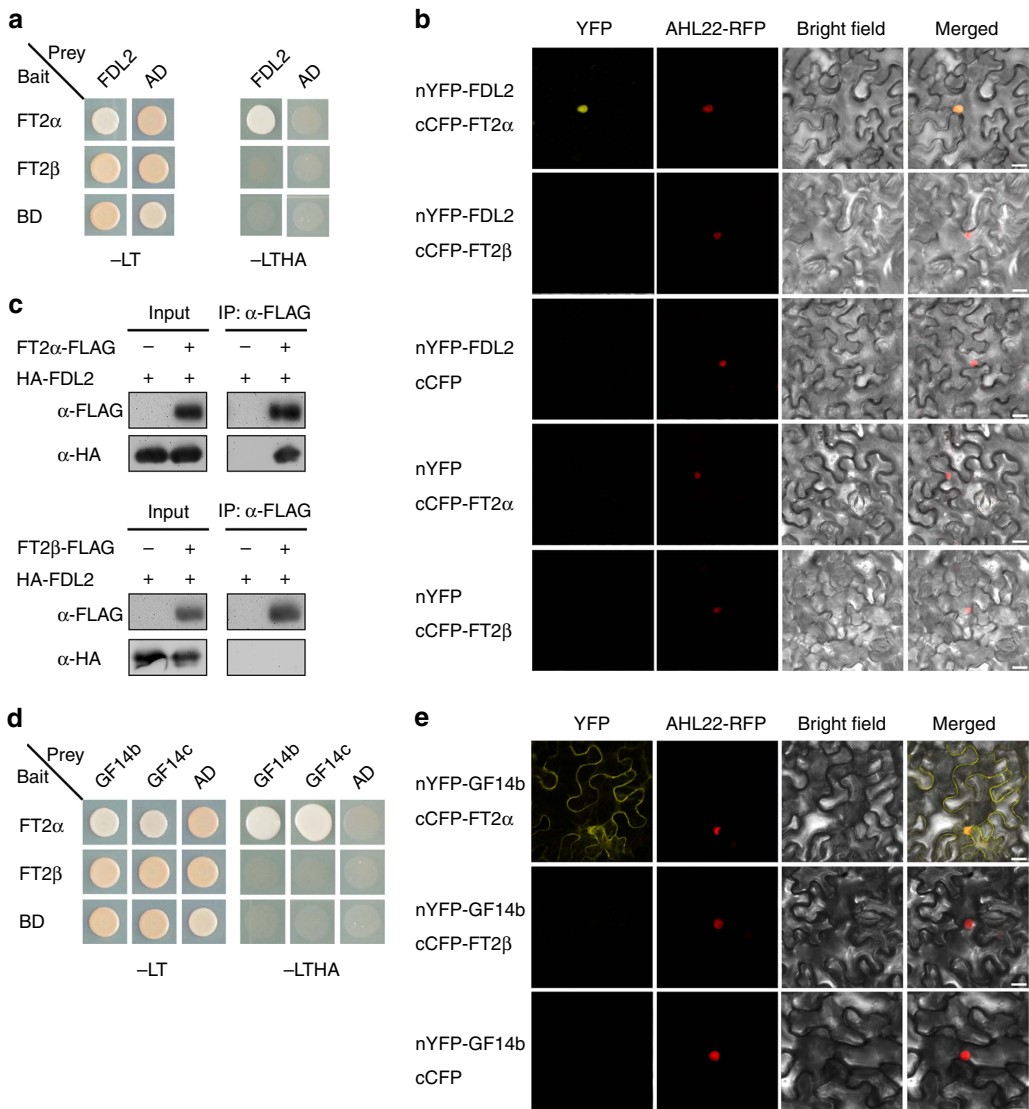

**Figure 2 | Interaction analysis of FT2α and FT2β with FD-like and 14-3-3 proteins.** (**a**) Interaction analysis of FT2α and FT2β with FD-like protein (FDL2) in yeast cells. Yeast cell growth on selective media without Leu, Trp, His and Ade ( − LTHA) indicates positive interactions. (**b**) BiFC interaction assays of FT2α and FT2β with FDL2. Partial YFP fusion constructs were expressed transiently in *N. benthamiana* leaves. AHL22-RFP was used as a nuclear marker. Scale bar, 20 μm. (**c**) *In vivo* Co-IP assays of two FT2 splicing isoforms with FDL2. FLAG-tag FT2α or FLAG-tag FT2β were co-expressed with HA-tag FDL2 in *N. benthamiana* leaves. (**d**) Yeast two-hybrid analyses of interactions between FT2 isoforms and two 14-3-3 proteins (GF14b and GF14c). (**e**) BiFC interaction assays of FT2α and FT2β with 14-3-3 protein GF14b. Scale bar, 20 μm.

relevance to temperature-mediated flowering, even though the *FT2β*/*FT2α* ratio can change in response to the environmental temperature.

Previously, we reported that *FT1* and *FT2* transcription level gradually increase with development under LD conditions in *B. distachyon*[14]. Thus we asked whether *FT2* AS changes with plant age. Using a qRT-PCR analysis with splicing-variant-specific amplification primers, the ratio of *FT2β*/*FT2α* abundance was found to decline during plant growth (Fig. 5a,b; Supplementary Figs 10 and 11). Although the transcription levels of *FT2α* and *FT2β* gradually increased throughout plant development as a result of photoperiodic induction, the proportions of different *FT2*-splicing isoforms changed with increasing plant age. *FT2β* accumulated to a much higher level than *FT2α* during the first 4 weeks, but *FT2α* rose much faster than *FT2β* afterwards (Fig. 5a; Supplementary Figs 10 and 11), thereby resulting in a gradual decrease in the *FT2β*/*FT2α* ratio as development progressed (Fig. 5b).

To further determine the effects of increasing age on *FT2* AS, we examined the diurnal expression rhythms of *FT2α* and *FT2β* in 4- and 7-week-old *B. distachyon* leaves. As shown in Fig. 5c and Supplementary Fig. 12, the *FT2β* expression level was higher than that of *FT2α* in 4-week-old plants during the day–night cycle. By contrast, more *FT2α* accumulated than *FT2β* in 7-week-old plants (Fig. 5d; Supplementary Fig. 12), further confirming the development-dependent change in *FT2* AS.

Therefore, in addition to the transcriptional control of *FT*, the posttranscriptional regulation of *FT* by AS may also contribute to flowering at the appropriate time in *B. distachyon*. We propose that a high level of *FT2β* in young plants may prevent precocious flowering, which allows plants to accumulate sufficient biomass to produce enough seeds to propagate the next generation.

**Down regulation of *FT2β* induces early flowering.** We previously found that the overexpression of miR5200, which

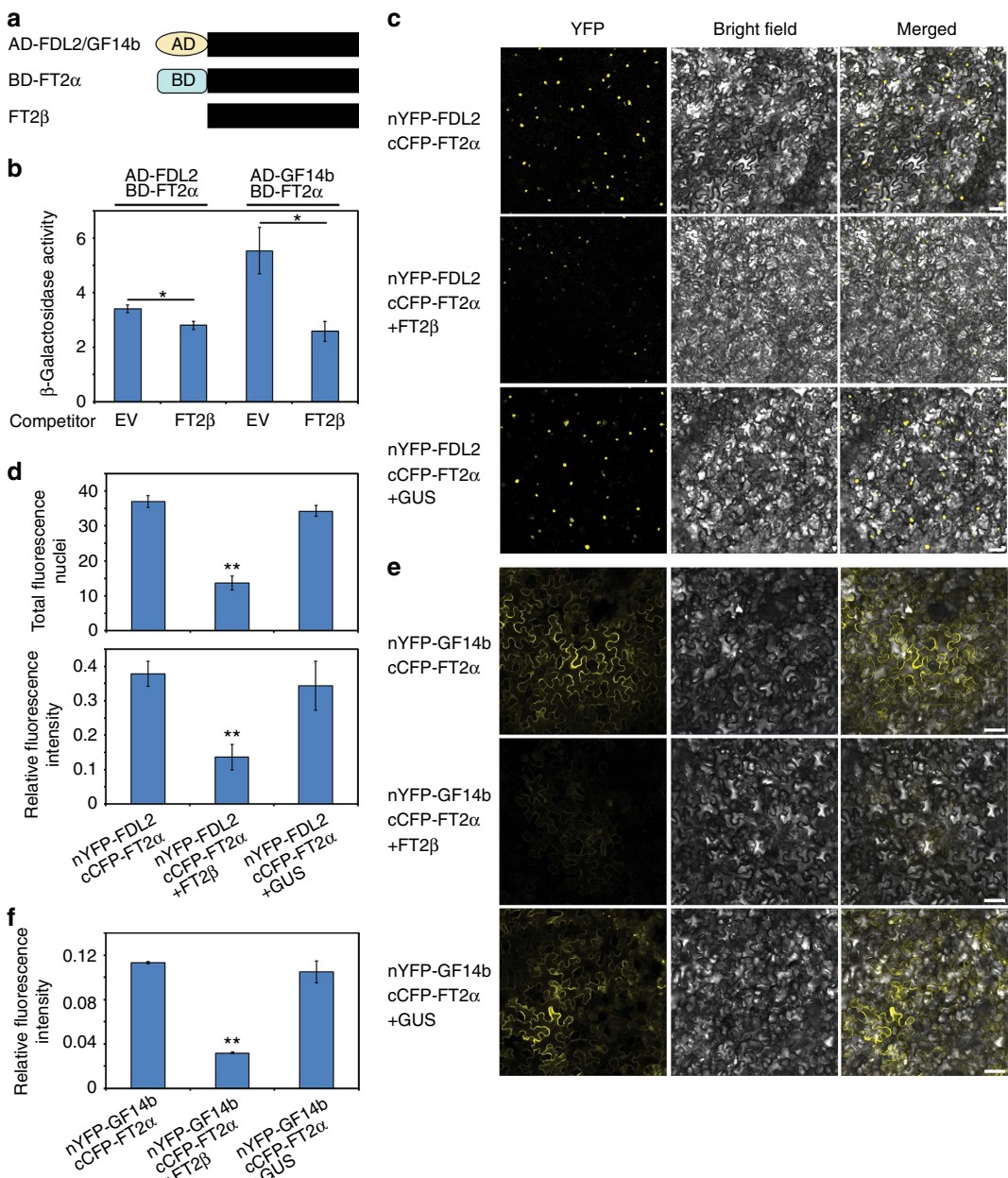

**Figure 3 | Attenuation of FT2α-binding capacity with FD-like and 14-3-3 proteins by FT2β.** (**a**) Constructs used for yeast three-hybrid assays. FT2β was used as the competitor to the interacting proteins with activation domain (AD) and DNA-binding domain (BD). (**b**) Relative quantification of inhibition activities of FT2β in yeast three-hybrid assays. β-Gal activities were measured in the presence or absence of FT2β (Student's *t*-test, *$P < 0.05$). Bars indicate s.e.m. of three replicates. EV, empty vector. (**c**) Representative photographs of fluorescence signals in BiFC assays for determination of FT2β effects on FT2α and FDL2 interactions in *N. benthamiana* leaf cells. GUS was used as a control in the absence of FT2β. Scale bar, 50 μm. (**d**) Total fluorescence nuclei and relative fluorescence intensity in BiFC assays used for analyses of FT2β inhibition activities to FT2α and FDL2 interactions. Bars indicate s.e.m. of three replicates (Student's *t*-test, **$P < 0.01$). (**e**) Representative photographs showing the fluorescence signals in BiFC assays for determination of FT2β effects on FT2α and 14-3-3 interactions in *N. benthamiana* leaf cells. GUS was used as a control in the absence of FT2β. Scale bar, 50 μm. (**f**) Relative fluorescence intensity in BiFC assays used for determination of FT2β-inhibitory activities to FT2α and 14-3-3 interactions. Bars indicate s.e.m. of three replicates (Student's *t*-test, **$P < 0.01$).

simultaneously targets both *FT1* and *FT2*, led to a severe delay of flowering onset in *B. distachyon*[14], suggesting that the suppression of *FT2α* expression may result in late flowering. To further characterize the role of FT2β in flowering control, we attempted to use artificial miRNAs to knock down *FT2β* expression, because artificial miRNA can specifically silence one splicing variant but not the other.

In plants, most of miRNAs start with a 5′uridine (U) for their correct sorting into ARGONAUTE 1 (AGO1) silencing

complex[44]. However, there is no adenine (A) nucleotide available for the design of artificial miRNA with a 5′U across the *FT2β* transcripts that skips the exon nucleotides of *FT2α* (Supplementary Fig. 13a). Even beginning with a 5′A, miR172 can be properly loaded into AGO1-silencing effectors as a result of its specific secondary structure[44]. Thus we designed an artificial miRNA using *MIR172a* as a backbone to downregulate *FT2β* (Fig. 6a; Supplementary Fig. 13a,b). We successfully obtained eight independent transgenic Bd21-3 plants and termed them

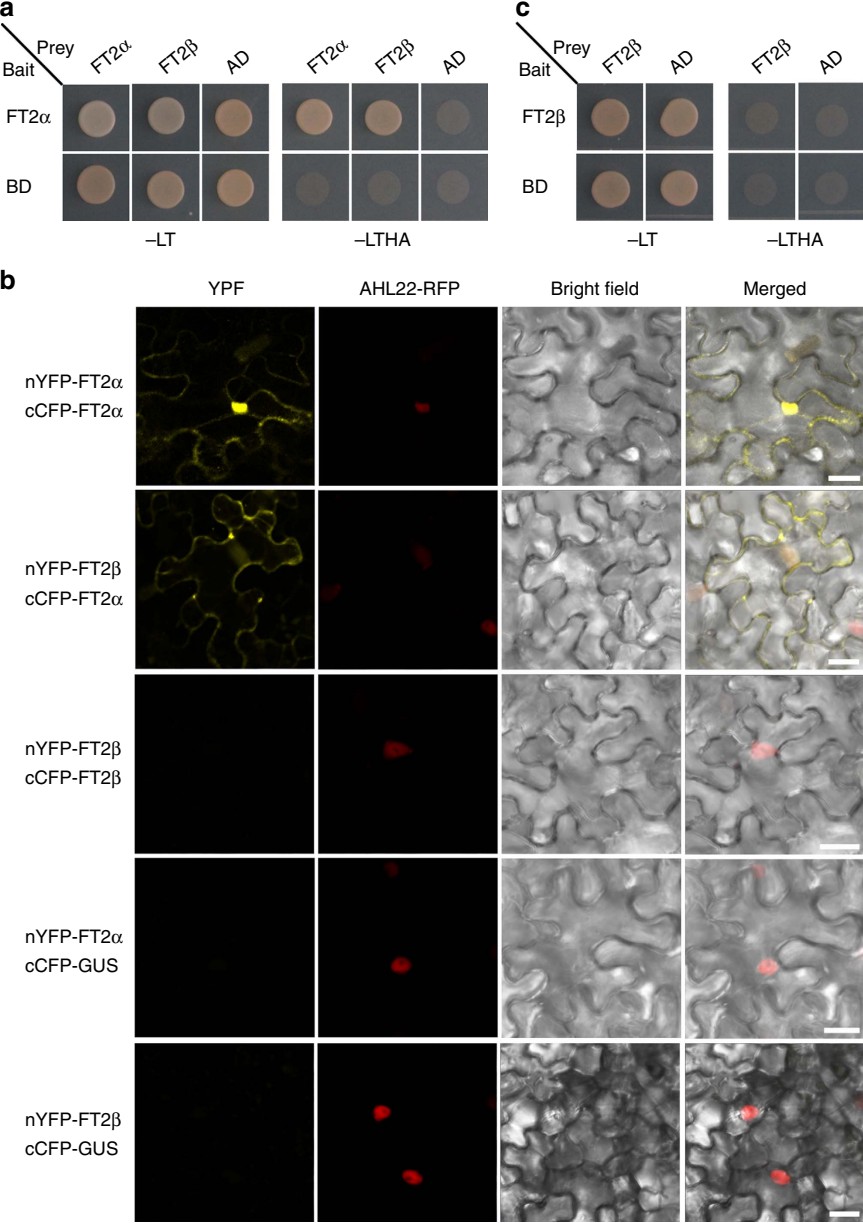

**Figure 4 | Homodimer and heterodimer formation analyses between FT2α and FT2β.** (**a**) FT2α can form homodimer and heterodimer with FT2β in yeast cells. (**b**) BiFC assays for FT2α and FT2β homodimer and heterodimer formation determination. GUS was used as a control in the absence of FT2α or FT2β. Scale bar, 20 μm. (**c**) FT2β cannot form homodimer in yeast cells. Yeast cell growth on selective media without Leu, Trp, His and Ade ( − LTHA) indicates positive interactions.

*amiRFT2β*. RNA analysis revealed high accumulation of artificial miRNA and are resultant decrease of *FT2β* expression in typical transgenic plants (Fig. 6b,c; Supplementary Fig. 14a,b). In addition,we did not find reductions in the gene expression levels of *FT2α* or other potential amiRFT2β targets in *amiRFT2β* plants (Fig. 6d; Supplementary Figs 14c,d and 15), suggesting that our artificial miRNA, which was introduced into plants, can specifically act on the desired transcripts.

An analysis of the flowering time of these transgenic plants, in the $T_3$ generation, under LD conditions revealed early heading (Fig. 6e). The heading date was earlier by an average of 8 days in *amiRFT2β* compared with those of control plants (54 versus 62 days, on average) (Fig. 6f). Consistent with these phenotypic observations, *VRN1* expression was two fold to four fold higher in *amiRFT2β* than in wild-type plants (Fig. 6g; Supplementary

Fig. 14e,f). The extent of early flowering phenotype in different *amiRFT2β* lines correlated with the mature artificial miRNA accumulation as well as the reduced *FT2β* expression, suggesting that the acceleration of flowering by reduced FT2β activity is likely due to the formation of more flowering activation complex and the resulting enhanced *VRN1* expression.

To help exclude the possibility that the early flowering in *amiRFT2β* is due to the unspecific effects of the artificial miRNA, we generated *amiRFT2α* transgenic lines, which specifically silenced *FT2α* expression by introducing a different artificial miRNA that targeted the intron region of *FT2α* where it is different from that of *FT2β* (Supplementary Fig. 13). We observed a severe delay of flowering (94 versus 68 days) and a reduction in *VRN1* expression in strong *amiRFT2α* lines (Supplementary Figs 16 and 17). Weak *amiRFT2α* lines exhibited

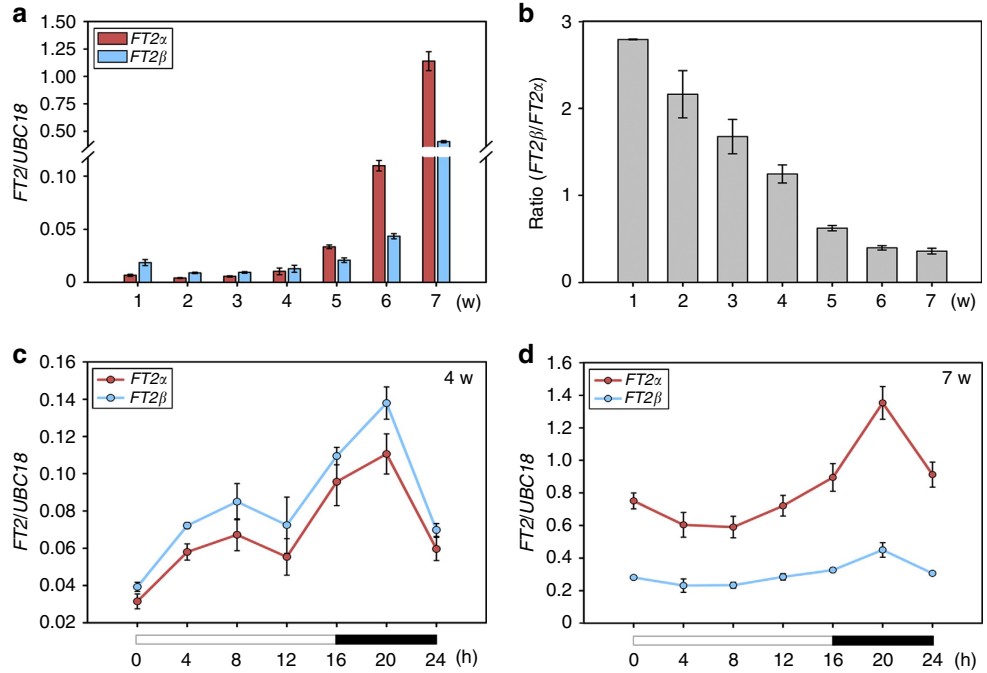

**Figure 5 | Age-dependent FT2 AS.** (**a**) Expression analysis of FT2α and FT2β along with different age. Upper flat leaves from five Bd21 plants with the indicated age were pooled and collected for RNA isolation. Each qRT-PCR analysis was performed in three biological replicates with similar results. The point represents the mean value of three technical replicates in a representative biological experiment. The transition from vegetative to reproductive stage of Bd21 plants occurs at 4–5 weeks. Standard curve for FT2α, FT2β and UBC18 quantification is shown in Supplementary Fig. 2. The absolute value of FT2α, FT2β and UBC18 is shown in Supplementary Fig. 10a. Error bars indicate s.d. w, weeks. (**b**) FT2β/FT2α ratio during plant development. (**c**) and (**d**) Diurnal expression analysis of FT2α and FT2β in 24-h period when plants were 4 and 7 weeks old. The absolute value of FT2α, FT2β and UBC18 at each time point is shown in Supplementary Fig. 12. Upper flat leaves from five Bd21 plants at the indicated time were pooled and collected for RNA isolation. The white and black bars along the horizontal axes represent light and dark periods, respectively. The numbers below the horizontal axes indicate the time in hours. Error bars indicate s.d. h, hours, w, weeks.

comparable heading dates with those of wild-type plants (Supplementary Figs 16 and 17). This may be because of the limited decrease in FT2α expression or the redundant activities of FT1 and FT2 in *B. distachyon*. The lack of early flowering phenotypes in all four *amiRFT2α* lines suggested that the precocious heading of *amiRFT2β* was due to the compromised FT2β activity rather than the unspecific effects of the artificial miRNA.

Taken together, the acceleration and delay in heading dates in *amiRFT2β* and a*miRFT2α* plants, respectively, further indicated that FT2β and FT2α have antagonistic roles in the flowering processes of *B. distachyon*.

**Dynamic change in FT2 AS is prevalent in temperate grasses.** In light of the opposite roles of FT2 splicing isoforms in *B. distachyon* flowering control demonstrated above, we asked whether the age-dependent modulation of FT2 AS was conserved among other plants. Using RT-PCR, similar splicing variants of FT2 orthologues in *A. thaliana*, rice and maize were not detected, even though the gene structures of the FT2 homologous genes in rice and maize are the same as that in *B. distachyon* (Supplementary Fig. 18).

As *B. distachyon* belongs to *Pooideae*, we investigated whether AS of FT2 is present in other temperate grasses. Indeed, we found two splicing variants of FT2 in *Aegilops tauschii*, wheat and barley, which differed by the loss of nucleotides that encode the same amino acids as in *B. distachyon* (Supplementary Fig. 19). Additionally, we observed that wheat and barley FT2β were expressed at much lower levels than FT2α in 8-week-old plants,

whereas FT2α transcripts were less abundant in 2-week-old plants (Fig. 7a,b). These results suggest that the change in the AS of FT2 regulated by development is conserved in temperate grasses.

## Discussion

AS is a common gene regulatory mechanism that greatly increases the complexity of transcripts and proteins encoded by an organism's genome. From high-throughput transcriptome studies in plants, it is now clear that AS is much more prevalent than previously estimated[37]. However, when, how and why these splicing variants are generated remains, to a large extent, unknown[22,23]. In this study, we reported that the highly conserved flowering integrator gene, *FT*, is also subjected to AS in temperate grasses. FT2β can heterodimerize with FT2α and FT1 in plant leaves and is unable to bind FD and 14-3-3s at the shoot apex, resulting in its interference with the formation of flowering inductive complexes. Thus it has a dominant-negative role in controlling flowering onset. The importance of change in FT2 AS during development was demonstrated by the alteration in heading date timing, which results from artificial-miRNA induced reduction in FT2β activity in *B. distachyon* (Fig. 6).

AS is thought to regulate functional gene actions in two ways: by directly altering mRNA stability or by encoding distinct protein isoforms, which may influence physical interactions between the original protein and its associates[23]. The latter is frequently associated with AS that is regulated by environmental stress or temperature response[24,25,28,45,46]. Although we detected that FT2β/FT2α ratio was influenced by ambient temperature

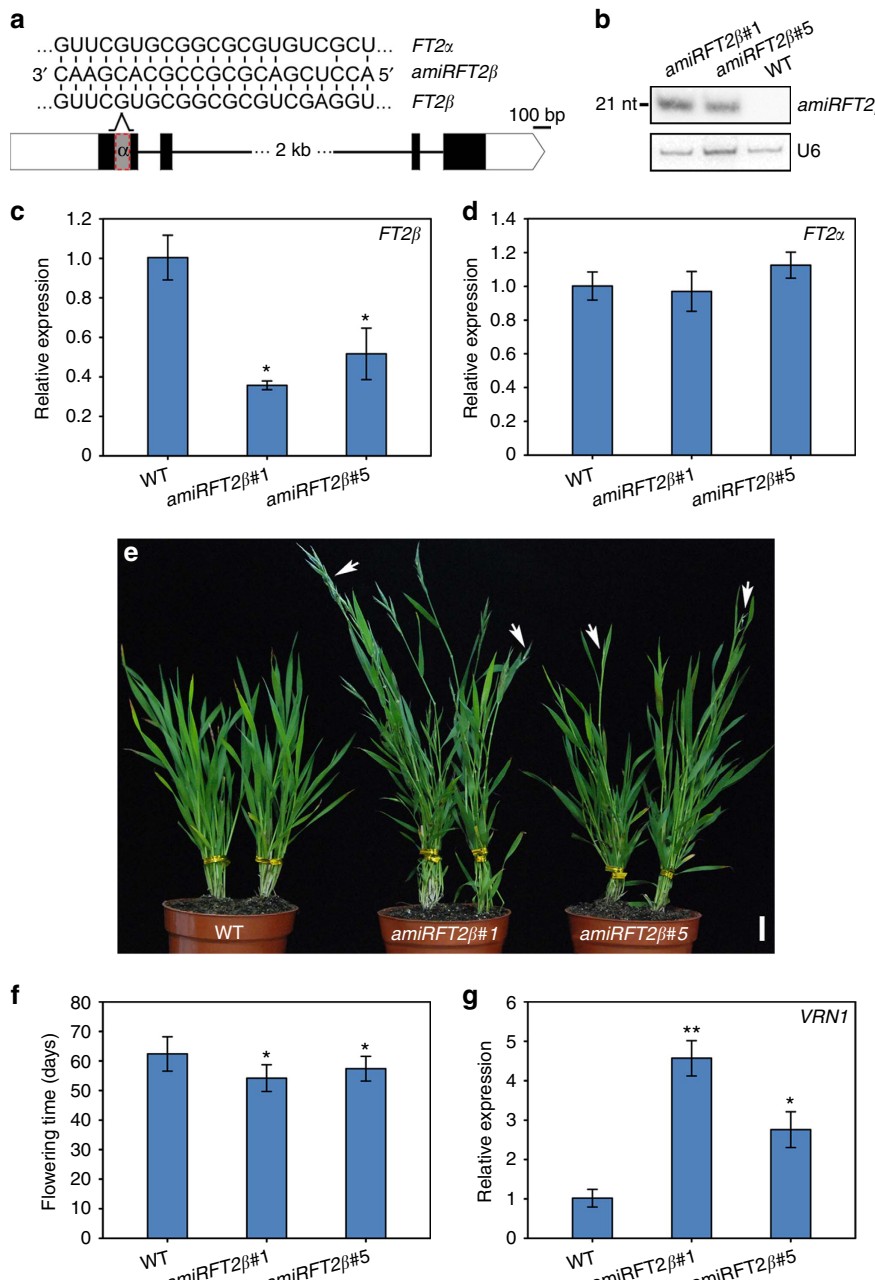

**Figure 6 | Interruption of FT2β activity by artificial miRNA results in early flowering.** (**a**) Schematic diagram of *amiRFT2β*. (**b**) Northern blotting analysis of artificial miRNA accumulation in *amiRFT2β* transgenic plants. U6 was used as a loading control for RNA gel blot. (**c**) qRT-PCR analysis of *FT2β* expression in wild-type Bd21-3 and the indicated *amiRFT2β* transgenic plants. *UBC18* was used as an internal control for normalization of qRT-PCR results. Three six-week-old *B. distachyon* plants were pooled and collected for RNA isolation. Each qRT-PCR analysis were performed in three biological replicates with similar results. The point represents the mean value of three technical replicates in a representative biological experiment. Error bars indicate s.d. Student's *t*-test, *$P < 0.05$ (**d**) qRT-PCR analysis of *FT2α* expression in wild-type and the indicated *amiRFT2β* transgenic plants. (**e**) Representative photograph of flowering phenotypes in *amiRFT2β* transgenic and wild-type Bd21-3 plants. White arrows point to spikes. Scale bar, 2 cm. (**f**) Flowering time of wild-type and the indicated two lines of *amiRFT2β* transgenic plants. Error bars indicate s.d. ($n = 10$). Student's *t*-test, *$P < 0.05$ (**g**) qRT-PCR analysis of flowering downstream gene *VRN1* expression in wild-type and the indicated *amiRFT2β* transgenic plants. Total RNA from 5-week-old whole-plant tissues, including leaves and shoot apex, were used for *VRN1* examination. Student's *t*-test, **$P < 0.01$, *$P < 0.05$.

changes, the AS of *FT2* may not be involved in temperature-mediated flowering control in *B. distachyon*, as increased ambient temperatures have limited influences on flowering signal induction in temperate grasses[43].

Interestingly, the *FT2β*/*FT2α* ratio was progressively reduced during development, indicating that *FT2* AS is regulated by an endogenous cue rather than an external cue for fine-tuning plant flowering. *FT2* AS may be modulated by some special splicing factors that abundance-, localization- or posttranslational modification-related changes as plant grow. Because miR156 and miR172 participate in the age-dependent regulation of flowering in diverse plants[12,13], it will be interesting to explore whether alterations in *FT2* growth-related AS in *B. distachyon* is controlled by these two miRNAs during flowering processes.

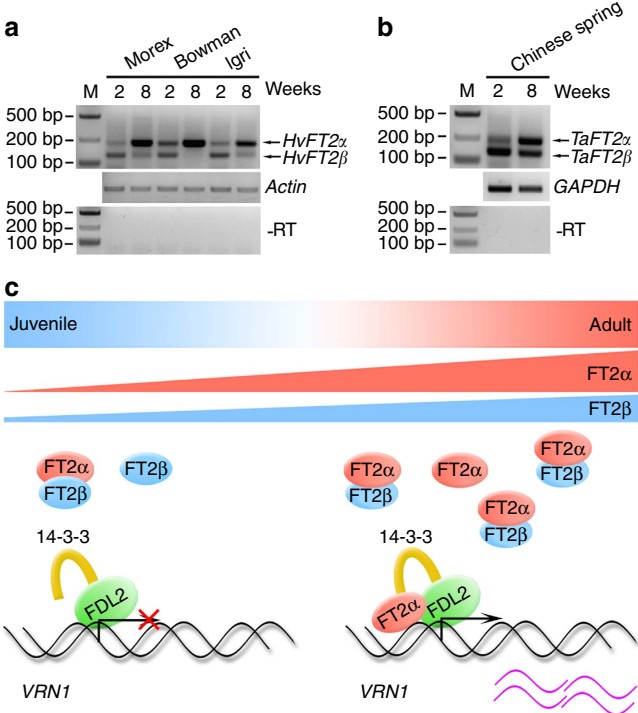

**Figure 7 | Dynamic changes in FT2 AS with plant development is prevalent in barley and wheat.** (**a**) RT-PCR analysis of FT2α and FT2β expression in barley with the indicated age. Primers designed for simultaneous amplification of FT2α and FT2β in a single PCR reaction were used. Total RNA from three typical varieties of barley was isolated for reverse transcription. M, marker. (**b**) RT-PCR analysis of FT2α and FT2β expression in wheat with different ages. Total RNA from Chinese Spring was isolated for reverse transcription. (**c**) A working model for two FT2 splicing isoforms during plant flowering-induction processes in temperate grasses. Although both two FT2 splicing transcripts increase along with plant development, their increasing rates are not consistent, resulting in distinct FT2β/FT2α ratio with different age. FT2β and FT2α can form heterodimers. Thus when plants are vegetative, the formation of flowering-induction complex is blocked, because FT2β expression level is higher than that of FT2α. When plants grow to reproductive stage, more FT2α accumulates than FT2β, and the flowering initiation complex can be formed in the SAM to induce VRN1 expression, ensuring eventual plant flowering at right growth phase.

FT2 and FT1 in *B. distachyon* are thought to have redundant roles on florigen activity because of their high amino-acid identities, similar temporal and spatial expression patterns, and same early flowering phenotype if overproduced. Thus it seems likely that FT2 and FT1 may have an analogous regulatory mechanism[14,15]. However, FT2 and FT1 are not regulated by identical schemes at the posttranscriptional level. First, their regulation by miR5200 is different. The base-pairing of miR5200 with the *FT2* target site is not as perfect as that of *FT1*, resulting in the cleavage of *FT2* by miR5200 at a non-canonical miRNA cleavage site[14]. Second, *FT2*'s function is modulated by AS, which is altered during development to repress flowering in the early stage, while no *FT1* AS events occur during the life cycle. Because both AS- and miRNA-mediated gene silencing are posttranscriptional gene regulatory mechanisms in eukaryotes, we propose that posttranscriptional modulation may be the major force for differentiating and fine-tuning different FT-like protein activities in *Pooideae* plants during evolution.

In *A. thaliana*, BROTHER OF FT AND TFL1 (BFT), a member of the FT/TFL1 protein family, mediates the link between flowering and stress signaling[47]. BFT is expressed at low levels, irrespective of day length, but can be dramatically induced under highly saline conditions. BFT and FT have similar abilities to bind FD. Thus, when plants encounter salt stress, BFT causes delayed flowering by competing with FT for FD binding[47]. Similarly, TFL1 represses FT activity through its interactions with FD, which is involved in the prevention of flowering in immature meristems of *A. thaliana*, *A. alpina* and even apple trees[48,49–52]. In *B. distachyon*, although developmentally regulated FT2β also interferes with flowering initiation complex formation, its molecular mechanism is different from that of BFT in *A. thaliana* because FT2β does not bind FD. In a sense, FT2β functions as a competitor of FD, because both FT2β and FD can dimerize with FT2α. FT2β/FT2α ratio is high in young plants and this prevents the rapid progression towards flowering. If there was no FT2β generated at the vegetative stage, then the florigen activity conferred by either FT2α or FT1 would immediately exceed the threshold for the reproductive phase transition, which may lead to precocious flowering and detrimental effects on seed production.

We found that full-length FT2 in *B. distachyon* can form homodimers and FT2-FT1 heterodimers. Similar findings that FT interacts with itself in *A. thaliana* suggest a possible biological relevance for the multimeric forms of FT proteins in plants[48]. Hence, further studies on when and where FT proteins form polymers *in vivo* will be important for determining their exact roles in accelerating flowering.

Plants contain complex regulatory machineries to optimize the seasonal timing of flowering. Our present discovery of AS-mediated *FT2* regulation in temperate grasses not only extends the manner of flowering control by FT but also reveals florigen control at both the transcriptional and posttranscriptional levels during the process of floral transition (Fig. 7c). In respect that multiple alternatively spliced forms of *FT* ortologous genes have been detected in *Platanus acerifolia*[53], it will be interesting to determine whether the blocking flowering complex formation by splice variants is the same *FT* regulatory mechanism in trees.

## Methods

**Plant materials and growth conditions.** *B. distachyon* used in this study were grown under LD conditions (16 h light/8 h dark) in growth chambers with temperatures of 22 °C during the day and 16 °C at night. For most experiments, the *B. distachyon* accession Bd21 was used. To minimize the crosstalk effects of photoperiod and vernalization on Bd21-3, seeds of transgenic and control plants were placed on moistened filter paper in Petri dishes and subjected to 4 °C cold treatment for 1 week before planting under the indicated conditions. Unless stated otherwise, all plant seedlings or leaves were harvested at Zeitgeber time 16 for use in the experiments.

**Constructs and plant transformation.** The *FT2* (*FT2α* and *FT2β*) coding sequences and the miR172a backbone of the artificial miRNA were amplified from *B. distachyon* cDNA. *FT2α*, *FT2β* and artificial miRNAs constructs were all under the control of the maize (*Zea mays*) UBIQUITIN (*UBI*) promoter for overexpression in Bd21 or Bd21-3 plants. The sequences of the primers for *B. distachyon FT2α*, *FT2β* and artificial miRNA amplification, as well as for binary vector construction, are listed in Supplementary Table 1. Transformation of *B. distachyon* was mediated by *Agrobacterium tumefaciens* strain AGL1 using compact embryogenic calli derived from immature embryos[54]. Independent transgenic lines were genotyped for T-DNA using *HPT II*-specific forward and reverse primers. The confirmed positive T₃ transgenic plants were used in flowering time determination and gene expression assays.

**Plant flowering time measurements.** The flowering times of the indicated plants were measured as the number of days from the date of planting in soil to the day when the first spike emerged. At least 10 plants of each transgenic line were recorded to calculate the average flowering time. Statistical significance was determined using Student's *t*-test.

**RNA expression analysis.** Total RNA was isolated from at least three pooled *B. distachyon* plants using TRIzol reagent (Invitrogen). The total RNA sample was pretreated with RNase-free DNase I (Promega) prior to cDNA synthesis to remove DNA and then reverse transcribed by EasyScript reverse transcriptase (TransGen Biotech) using oligo (dT) primers. RT-PCR analyses of gene transcripts were essentially carried out as previously described[55]. qRT-PCR reactions were performed with SYBR Premix EX Taq (Takara) and run on an ABI 7,500 Fast system with the following procedure: 95 °C for 1 min, then 40 cycles of 95 °C for 10 s, 62 °C for 10 s, and 72 °C for 30 s, followed by a melting dissociation program.

For the quantification of *FT2α* and *FT2β* transcripts, *FT2α* and *FT2β* cDNAs were cloned into pGADT7 vectors, and an absolute standard curve of each transcript was made using 10-fold serial dilutions from $10^{-18}$ to $10^{-23}$ mol μl$^{-1}$. The gene expression levels (Fig. 5 and Supplementary Fig. 3) were calculated based on the ratio of absolute quantifications of *FT2α* and *FT2β* to the corresponding absolute value of *UBC18*. For gene relative expression analysis, *UBC18* mRNA was examined in parallel and used for data normalization, and the relative expression levels were calculated using the ΔΔCt method.

All of the qRT-PCR analyses were performed at least in three independent biological replicates with three technical repetitions. For artificial miRNA detection, 2 μg total RNA was used for RNA gel blots. $^{32}$P-end-labelled oligonucleotides complementary to the artificial miRNA sequences were used as probes. The sequences of the primers and small RNA blot probes are listed in Supplementary Table 1.

**Yeast assays.** Yeast two-hybrid assays were conducted using the Matchmaker GAL4 Two-Hybrid System (Clontech). The PCR products of *FT2α*, *FT2β*, *FT1*, *FDL2* and diverse *14-3-3s* from *B. distachyon* cDNA were independently cloned into the pGBKT7 and pGADT7 vectors. Transformation of AH109 cells was performed according to the general protocol. The transformed yeast cells were spread on the selective medium SD/-Leu/-Trp/-His/-Ade plus 2.5 mM 3-amino-triazole to determine the interactions.

For the yeast three-hybrid assay, bait and prey were co-transformed with or without the competitor *FT2β* into yeast AH109 cells. The interaction strengths were quantified by a liquid β-Gal assay using chlorophenol red-β-D-galactopyranoside (CPRG) as substrates according to the Yeast Protocol Handbook (Clontech). Briefly, the yeast cells were broken open using several freeze/thaw cycles and mixed with CPRG buffer. When the colour of the sample turned to red, 3.0 mM ZnCl$_2$ was added to stop the reaction. The β-Gal units were calculated by the supernatant absorbance at 578 and 600 nm.

**BiFC assays.** To generate the constructs for the BiFC assays, full-length cDNA fragments of *FT2α*, *FT2β*, *FT1*, *FDL2* and *GF14b* were amplified and cloned into the pDONRZeo (Invitrogen) vector for fusion with the N-terminus of YFP or the C-terminus of CFP by LR reaction. The resulting plasmids were introduced into *A. tumefaciens* EHA105, which were then infiltrated into 4-week-old *N. benthamiana* leaves for transient expression. In brief, the *A. tumefaciens* strains were resuspended to OD600 = 0.8 in infiltration buffer (150 μM acetosyringone, 10 mM MgCl$_2$ and 10 mM MES). The resuspensions containing each BiFC construct as well as the p19 plasmid were mixed at 1:1:1 ratio and then infiltrated into leaves with a syringe.

For the protein-binding competition assay, the *A. tumefaciens* harbouring the *FT2β* plasmid was co-infiltrated with an equal volume of the indicated construct. *A. tumefaciens* harbouring *GUS* in place of *FT2β* was infiltrated in parallel as a control. The tobacco leaves were imaged under Zeiss LSM 700 confocal microscope 3 days after infiltration. Excitation of YFP was performed using an argon laser line at 514 nm and red fluorescent protein (RFP) with 563 nm, while capture of YFP fluorescence was at 518–555 nm and RFP at 568–636 nm. Primers used to construct BiFC plasmids are shown in Supplementary Table 1.

**Co-IP assays.** Full-length coding sequences of *FT2α*, *FT2β* and *FDL2* were cloned into tagging plasmids under the control of Cauliflower Mosaic Virus 35S promoter. The 4-week-old *N. benthamiana* leaves were infiltrated with an *A. tumefaciens* strains harbouring FT2α-FLAG plus HA-FDL2 or FT2β-FLAG plus HA-FDL2. The infected leaves were collected 3 days after infiltration. Total protein was then extracted with IP buffer (50 mM Tris-HCl pH 7.5, 150 mM NaCl, 1 mM EDTA, 1% Triton X-100, 10% glycerol, 1 mM phenylmethyl sulfonyl fluoride, 1 mM NaVO$_4$, 50 μM MG132 and 1× protease inhibitor cocktail) and centrifuged three times at 16,000*g* at 4 °C. The supernatant was incubated with Anti-Flag M2 Affinity Gel (Sigma, A2220) at 4 °C for 3 h with gentle shaking. After centrifugation at 600*g* at 4 °C for 1 min, the agarose beads were washed five times with washing buffer (50 mM Tris-HCl pH 7.5, 150 mM NaCl, 1 mM EDTA, 0.1% Triton X-100). The precipitate was suspended, boiled in SDS loading buffer, separated by an SDS-PAGE and detected by anti-HA (Sigma, H6908, 1:5,000 dilution) or anti-FLAG antibody (Sigma, H7425, 1:3,000 dilution). The full scan picture of Co-IP assay results is shown in Supplementary Fig. 20.

**Data availability.** The sequence of *FT2β* has been submitted to Genbank under accession code (KP637176). The authors declare that all other data that support the findings of this study are available in the manuscript and its Supplementary files or are available from the corresponding author upon request.

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

## Acknowledgements

We sincerely thank Dr Daniel Klessig for reading the manuscript and improving writing. This work was supported by the National Natural Science Foundation of China (91640109, 31570226, 31200181), Innovation Program Chinese Academy of Agricultural Sciences and Central Government Non-profit Institutional Grants, Zhejiang Provincial Natural Science Foundation of China (LR16C060001), the Fundamental Research Funds for the Central Universities (2016QNA6014) and Hundred-Talent Program of Zhejiang University.

## Author contributions

L.W. and L.M. conceived and supervised the research. L.W., L.M., Z.Q. and J.W. designed the experiments. Z.Q., J.W., G.S., F.C., N.F., X.K., S.G., K.C. and L.W. performed the experiments. Z.Q., J.W., A.L., L.M. and L.W. analyzed the data. L.W., L.M. and Z.Q. wrote the article.

## Additional information

**Competing financial interests:** The authors declare no competing financial interests.

**Publisher's note**: 
 

