## [Peer Review File · Nature Communications]

Reviewers' comments:

Reviewer #1 (Remarks to the Author):

The manuscript by Qin et al. demonstrates the role of alternative splicing (AS) in one of the orthologs (FT2) of the gene encoding the florigen protein that controls the change from vegetative to reproductive shoot development in plants. They focus their studies in the temperate grass *Brachypodium distachyon* as a model for the behavior of other grasses with more agricultural interest such as wheat and barley. The main findings of the paper are that FT2 generates 2 AS isoforms differing by a 28 amino acid segment. The longest, full length, isoform (FT2 alpha) is fully functional. However, the shorter FT2beta cannot interact with the physiological partners FD and 14-3-3s, but is able to form heterodimers with FT2alpha and FT1, thereby interfering with the florigen-mediated assembly of the flowering initiation complex. While overexpression of FT2beta delays flowering, artificial miRNA silencing of FT2beta causes accelerated flowering. Overall, the results demonstrate that FT2beta acts as a dominant negative of FT2alpha, and therefore prevents or inhibits its flowering promoting role.

In general, the experimental approaches of this manuscript are varied, solid and use state of the art methodologies in the field. These include yeast two hybrid studies, reconstitution of functionality with hemi-proteins, overexpression and silencing of proteins in transgenic plants and AS assessment. Experiments are well controlled and I do not find overinterpretations. Nevertheless, the fact that the functionality of a master transcription factor is controlled by an AS mechanism in which one of the splicing variants acts as a dominant negative is not novel. It has been profusely demonstrated both in animal cells and plants, the latter cases being cited by the authors.

In the absence of novelty in the found mechanism, what I find more interesting is the less developed part of the paper dealing with the bases for the splicing regulation and the functional implications of this mechanism. These include two important aspects to investigate in more depth:

1. The finding that the FT2beta/FT2alpha ratio decreases with time in a way that it may perfectly explain the physiological need to delay flowering until the plant biomass has achieved a certain level (Figure 5). One would like to know here what are the molecular bases for the AS regulation that causes this ratio change as the plant grows. There is no need of a detailed mechanism here, but at least some hints on the splicing factors that regulate FT2 AS and whether their abundance, localization or post-translational modifications changes along plant growth.

2. The finding that wheat and barley may have the same mechanism described in *Brachypodium distachyon*. In this case, the authors only mention lack of conservation in *Arabidopsis* and provide partial evidence of conservation in wheat and barley. A more extensive study encompassing several dicot and monocot species of agricultural interest would shed light to know if the found mechanism is characteristic, prevalent or restricted to Poaceae.

In summary, in the present state, I feel that this manuscript is better suited for a more specialized journal. If the authors manage to provide new data along the lines mentioned in 1 and 2, the manuscript would acquire much more substance and become an excellent candidate for Nature Communications.

Reviewer #2 (Remarks to the Author):

The results presented here suggest that FT2 α promotes flowering while a shorter alternative splicing form FT2 β inhibits flowering in *B. distachyon*. The evidence of 2 alternative splicing forms of FT2 is interesting. However, its relevance to flowering in natural conditions is not conclusively demonstrated.

The overall effect of FT2 on flowering is not well characterized (only OE from previous study). The authors showed before that over-expression of FT2 in *Brachypodium* accelerates flowering so it acts as a promoter of flowering. FT2 RNAi in *Brachypodium* and FT2 null-mutants exhibit delayed flowering (unpublished data) confirming that FT2 acts as an overall promoter of flowering in these species. Therefore the short alternative-splice form and its putative repressive role seem to be overridden by the positive effect of FT2 on flowering. If the alternative short variant repression is biologically relevant, then the FT2 mutants should be early flowering, but the opposite is observed. The authors need to characterize better the role of FT2 gene (both forms together) on flowering.

The over-expression approach generates unusually high levels of this variant (9-fold higher than normal) that may never be present in the normal plant..., so its role at normal levels in flowering is still not clear. In spite of the 9-fold higher expression the effects are small (5 days in one transgenic and 11 days in the other one). I think that the conclusion that high levels of FT2 β in young plants may prevent precocious flowering is premature. Other mechanisms are involved in the prevention of early flowering.

There are new published data that shows that FT2 is expressed in the apex and that therefore it may not have to move through the phloem to produce an effect. The authors need to show the level and time of expression of the two FT2 variants in the apex.

The authors used an artificial miRNA to specifically silence the FT2 β variant. The idea is nice, and the result seems to agree with their model. However I think that they need extra evidence, as artificial miRNA could be unspecific. Also, there could be a problem with the design of the artificial miRNA. Almost always miRNAs start with a U at the 5' and this is very important for the miRNA loading in the proper silencing complex. Their artificial miRNA starts with an A, and this could send the miRNA to a different silencing complex. I know that they used the miR172 backbone, and miR172 is a particular miRNA that starts with A. However, the miR172 loading depends also in the secondary structure of the miRNA, and the authors need to explain clearly this potential pitfall. The effect on flowering is small (8 d) relative to the standard deviations presented in Figure 6F. The differences seem marginally significant (*). It would be better to increase the sample size to show a more conclusive effect on flowering

Interactions: the authors show that FT2 β does not bind to FDL2 and GF14b and GF14c (FT2 α does interact). In Y3H FT2 β reduces the strength of the interaction between FT2 α and

FDL2 and FT2 α and GF14. It is not clear to me how FT2 β interferes with the formation of the FAC if it does not bind to the GF14b and GF14c proteins that may mediate the interaction with FT2 α ... The authors indicate that this may be the result of direct interactions between FT2 β and FT2 α and FT1. The authors need to eliminate the possibility of indirect interactions mediated by the yeast 14-3-3 or the Nicotiana 14-3-3.

Test binding of FT2 β to yeast 14-3-3. It is possible that FT2 β binds to any of the other multiple 14-3-3 proteins in *Brachypodium* and by the FAC complexes using these 14-3-3? Please expand the set of *Brachypodium* 14-3-3 tested with the two alternative splice forms of FT2.

Reviewer #3 (Remarks to the Author):

The manuscript by Qin et al describes the characterization of a splice variant of a FLOWERING LOCUS T homologue in *Brachypodium distachyon*. The authors present two independent datasets on protein interaction that confirm that this splice variant, which they call FT2beta, encodes for a 5' internal deletion in the corresponding protein that makes it incapable of interacting with the *B. distachyon* proteins corresponding to previously characterized binding partners 14-3-3 and FD from *Arabidopsis thaliana*. In contrast, FT2alpha (the full-length version) and a second paralog can interact with both partners. Interestingly, FT2beta is not compromised in its capability to form heterodimers with FT2alpha or FT1. Furthermore, the authors show data of plants overexpressing ubiquitously either FT2beta or an artificial microRNA directed against FT2beta. These data show a clear and opposite effect on heading date for both scenarios. The authors propose as model that the truncated FT2beta version may act as competitive inhibitor to FT1 and FT2alpha, which act as florigen.

The authors also interrogate expression of both splice variants during development and observe that the ratios between FT2 beta and alpha change with age so that the more active FT2alpha predominates in older plants. These data seem plausible but the authors make the common mistake of over interpreting real-time qRT PCR data that are obtained with a relative quantification method. Therefore, some formulations have to be tuned down or, preferably, absolute quantifications of both transcripts need to be carried out (see detail below as major point).

Interestingly, the alternative splicing of FT2 seems conserved among temperate grasses. It is important to stress in the discussion that this means that they are potentially regulating but that first, this needs to be shown in the other species as well.

The discussion is rather obvious and does not open up towards as yet unsolved future questions, e.g. in which tissue the competitive mechanisms is mainly located.

Major points:

1. Page 7, second paragraph, third paragraph. Also page 8, third paragraph. All claims of „Gene A is expressed more than Gene B" are not supported by the method that seems to have been employed for quantification (not entirely detailed in methods). If no absolute quantification is used, qRT-PCR is a relative measure thus one control sample per primer pair is arbitrarily scaled to "1". Alternatively, the values are scaled to the average, but it is still an arbitrary scale. It is possible to say "gene A is more expressed in sample x as

compared to a control", but not "A more/less than B". This also concerns Figure 5A, where the reference point is not obvious for FT2alpha and FT2beta. In Figure 5B, the ratio changes, that is true, but the „1" line is randomly set. It is still possible that FT2beta is expressed at negligible levels throughout development. Figure 5C and D have the same problem as A. The authors seem to be aware of how to calculate qRT values, because Figure 6C,D and G show a control (WT) sample as clear reference point (set to „1" in this case).

2. All protein interaction assays involving the truncated „negative" FT2beta protein. Theoretically, absence of interaction can be explained by an unstable protein (in both assays). Unfortunately in the case of the split YFP, the amount of protein that accumulates is a major source of artefacts and negative controls should be expressed at similar levels and in the same compartments to make that assay at least a little reliable. Protein stability could be visualized by showing comparative expression levels for plan YFP fusion of all proteins under study. Alternatively, the pull-down input, which is basically produced in the same expression system, could be shown as control across one western (now it looks as if FT2alpha accumulated more than FT2beta but that is probably due to the exposure time at different blots). It is true that the positive interaction detected between the truncated FT and the full-length FTs indicates that this protein does indeed accumulate to adequate levels, making at least a false negative unlikely.

3. Figure 4B. For the reasons detailed above, the „empty vector" control is useless as the fusion protein does not accumulate at levels comparable to fusion proteins.

Minor points

1. The English needs to be fixed throughout the manuscript. Smaller changes, but too much for a peer review.

2. Abstract. FT1 and FT2 are not orthologs of FT. They are paralogs and homologs of FT. Ortholog should be reserved for genes with conserved synteny.

3. Abstract. Revise text for precision. E.g. splice variants do not interact, the encoded proteins do. Last sentence: it is not a regulation machinery but a mode of regulation that was discovered; the data do not apply to "FT in plants" but to FT2 in *B. distachyon* and possibly other temperate grasses (last sentence).

4. Citation 14 makes no sense in the context of FT cis regulation

5. The authors cite many reviews from the same groups (recent and older), which may be a little redundant. In contrast several key papers on FT regulation are missing.

6. AP2-like proteins do not bind in the 3'untranslated regions, but SMZ appears to bind downstream of FT based on the cited manuscripts.

7. *Arabidopsis* in italics would mean the entire genus. If *Arabidopsis thaliana* aka *Arabidopsis* is meant, do not use italics but introduce as abbreviated term.

8. Not clear to me why the renaming of FTL1 and FT2 and of FTL2 as FT1 is less confusing because of VRN3 in wheat and barley. It seems to yet add another layer of confusion to rename these genes again (page 3).

9. Page 6 „using transient GUS protein as control" is confusing. Better „As control, GUS instead of FT2beta was expressed.. „ or something similar.

10. Figure 1C. It would have been nice to see experimental data confirming that both splice variants use the same transcription start site and therefore also the same ATG (5'race). Since this has not been shown, indicate that in the legend - putative TSS for both transcripts.

11. Methods: RNA expression analysis: not clear how the analysis was done from this

description (deltadelta CT? What was the reference point? The data show SD of n=3?)

12. I do not find the „Bright field" and „merged" pictures for Figure 2B,E and 3C and 4B all that informativ

Responses to Reviewers' Comments

We wish to express our deep appreciation for the reviewers' constructive comments. We have performed additional experiments and revised the manuscript. We hope we have addressed all the concerns made by the reviewers. The point-by-point responses to the reviewers' comments are listed in detail below:

Reviewer #1

This reviewer said our experiments are varied, solid and are well controlled and not over interpreted and commented that "the manuscript would acquire much more substance and become an excellent candidate for Nature Communications". We have addressed the reviewer's comments in detail below.

(From the referee) The manuscript by Qin et al. demonstrates the role of alternative splicing (AS) in one of the orthologs (FT2) of the gene encoding the florigen protein that controls the change from vegetative to reproductive shoot development in plants. They focus their studies in the temperate grass *Brachypodium distachyon* as a model for the behavior of other grasses with more agricultural interest such as wheat and barley. The main findings of the paper are that FT2 generates 2 AS isoforms differing by a 28 amino acid segment. The longest, full length, isoform (FT2 alpha) is fully functional. However, the shorter FT2beta cannot interact with the physiological partners FD and 14-3-3s, but is able to form heterodimers with FT2alpha and FT1, thereby interfering with the florigen-mediated assembly of the flowering initiation complex. While overexpression of FT2beta delays flowering, artificial miRNA silencing of FT2beta causes accelerated flowering. Overall, the results demonstrate that FT2beta acts as a dominant negative of FT2alpha, and therefore prevents or inhibits its flowering promoting role.

In general, the experimental approaches of this manuscript are varied, solid and use state of the art methodologies in the field. These include yeast two hybrid studies, reconstitution of functionality with hemi-proteins, overexpression and silencing of proteins in transgenic plants and AS assessment. Experiments are well controlled and I do not find overinterpretations. Nevertheless, the fact that the functionality of a master transcription factor is controlled by an AS mechanism in which one of the

splicing variants acts as a dominant negative is not novel. It has been profusely demonstrated both in animal cells and plants, the latter cases being cited by the authors.

In the absence of novelty in the found mechanism, what I find more interesting is the less developed part of the paper dealing with the bases for the splicing regulation and the functional implications of this mechanism. These include two important aspects to investigate in more depth:

1. The finding that the FT2 β /FT2 α ratio decreases with time in a way that it may perfectly explain the physiological need to delay flowering until the plant biomass has achieved a certain level (Figure 5). One would like to know here what are the molecular bases for the AS regulation that causes this ratio change as the plant grows. There is no need of a detailed mechanism here, but at least some hints on the splicing factors that regulate FT2 AS and whether their abundance, localization or post-translational modifications changes along plant growth.

Response: In this manuscript, we found that the ratio of *FT2 β /FT2 α* is regulated by plant age, which is a new concept about AS regulation mode in plants, and further genetically and molecularly demonstrated that the endogenous cue-regulated *FT2* AS is of biological significance. We totally agree that the reviewer's comments on "the molecular bases for the FT2 AS regulation that causes this ratio change as the plant grows is interesting and One would like to know here what are the molecular bases for the AS regulation that causes this ratio change as the plant grows". Even if it is out of the scope of this manuscript and would be another story; we added some new experimental data as well as some discussions to provide more hints about the potential splicing factors control age-dependent *FT2* AS in the revised manuscript (see below).

In *Arabidopsis*, the splicing factor, *SCL33* (At1g55310), functions auto-regulates its alternative splicing¹. A recent paper reported that AS of *SCL33* orthologous gene in *B. distachyon* is dynamically regulated during plant development and during viral disease progression^{2,3}. We examined the expression of *SCL33* variants in 2-6 weeks *B. distachyon* plants and found that both the transcription as well as AS of *SCL33* are altered with different plant age in *B. distachyon* (Figure 1A below). Since different *SCL33* variants encode

proteins with divergent domains (Figure 1B below), we propose that developmentally-regulated *SCL33* may be involved in regulation of age-dependent *FT2* AS. We added these data in the revised discussion section.

Figure 1 Developmental regulation of *SCL33* transcription and splicing

(A) RT-PCR analysis of *SCL33* expression in *B. distachyon* under different age. Primers designed for simultaneous amplification of diverse *SCL33* splicing variants in a single PCR reaction was used. (*) and the numbers indicate the predominant transcripts of multiple *SCL33* splice variants. M, marker.

(B) Schematic structures of proteins encoded by predominant transcripts of *SCL33* splice variants. Some isolates have premature termination codons in their ORF, leading to encode truncated *SCL33* proteins without functional RRM and SR domains. Red star represents stop codon.

(From the referee) 2. The finding that wheat and barley may have the same mechanism described in *Brachypodium distachyon*. In this case, the authors only mention lack of conservation in *Arabidopsis* and provide partial evidence of conservation in wheat and barley. A more extensive study encompassing several dicot and monocot species of agricultural interest would shed light to know if the found

mechanism is characteristic, prevalent or restricted to Poaceae.

In summary, in the present state, I feel that this manuscript is better suited for a more specialized journal.

If the authors manage to provide new data along the lines mentioned in 1 and 2, the manuscript would acquire much more substance and become an excellent candidate for Nature Communications.

Response: In addition to Arabidopsis, rice, barley and wheat, we also examined the AS events of *FT2* homologous genes in maize, a major plant in Poaceae, as well as *A. tauschii*, a typical temperate grass in the additional experiments (Figure 7, Supplemental Figure 9 and Supplemental Figure 10). Even the *FT2* ortholog in rice and maize have the same gene structures (supplemental Figure 9A and 9B), we did not find splicing variants of them existed in rice and maize (supplemental Figure 9C), thus we hypothesized that *FT2* splicing mechanism may not be universal in all Poaceae plants. Because *Brachypodium*, barley, wheat and *A. tauschii* are typical plants of temperate grasses, and the same *FT2* splicing variants occur in all of them (supplemental Figure 10). Moreover, we found that these *FT2* AS are altered with plant growth stage (Figure 7), thus we got the conclusion that change of *FT2* AS along with plant development is prevalent in temperate grasses. Please see the revised results section.

Reviewer #2

The major concern from the review is about the biological relevance of *FT2* AS to flowering. After performing additional experiments, we think we can address this reviewer's concerns by providing new data.

(From the referee) The results presented here suggest that *FT2* α promotes flowering while a shorter alternative splicing form *FT2* β inhibits flowering in *B. distachyon*. The evidence of 2 alternative splicing forms of *FT2* is interesting. However, its relevance to flowering in natural conditions is not conclusively demonstrated.

The overall effect of *FT2* on flowering is not well characterized (only OE from previous study). The authors showed before that over-expression of *FT2* in *Brachypodium* accelerates flowering so it acts as a

promoter of flowering. FT2 RNAi in *Brachypodium* and FT2 null-mutants exhibit delayed flowering (unpublished data) confirming that FT2 acts as an overall promoter of flowering in these species. Therefore the short alternative-splice form and its putative repressive role seem to be overridden by the positive effect of FT2 on flowering. If the alternative short variant repression is biologically relevant, then the FT2 mutants should be early flowering, but the opposite is observed. The authors need to characterize better the role of FT2 gene (both forms together) on flowering.

Response: We are sorry that as far as we know, neither single mutant of *FT2* nor specific *FT2* RNAi lines have been obtained in *Brachypodium* so far. On the other hand, even if the *FT2* null-mutant exhibits the delay flowering, it is still difficult to exclude the negative role of FT2 β in flowering control because both *FT2 α* and *FT2 β* cannot express in a *FT2* null-mutant. To address this issue, we produced transgenic *Brachypodium* plants with artificial miRNAs specific targeting *FT2 β* and found that *amiRFT2 β* obviously accelerated the plant heading date and increased *VRN1* expression (Figure 6). Furthermore, to exclude the unspecific actions of artificial miRNA, in our additional experiments, we generated *Brachypodium* transgenic plants with an artificial miRNA specific targeting *FT2 α* . We observed that strong *amiRFT2 α* lines significantly delayed the flowering time with lower expression levels of *VRN1* compared with wild-type plants (Supplemental Figure 8). These genetic results strongly suggest that FT2 α and FT2 β play positive and repressive roles in flowering, respectively.

(From the referee) The over-expression approach generates unusually high levels of this variant (9-fold higher than normal) that may never be present in the normal plant..., so its role at normal levels in flowering is still not clear. In spite of the 9-fold higher expression the effects are small (5 days in one transgenic and 11 days in the other one). I think that the conclusion that high levels of FT2 β in young plants may prevent precocious flowering is premature. Other mechanisms are involved in the prevention of early flowering.

Response: We agree with the reviewer that overexpression plants may be premature to present the gene functions in the normal plants. In addition to

overexpression plants, we also used both *FT2* splicing variants silencing plants to observe their flowering performances. *FT2 α* and *FT2 β* silencing plants by artificial miRNAs lead to late and not late flowering, respectively, strongly suggesting that *FT2 α* and *FT2 β* play opposite roles in flowering in normal conditions (Figure 6) (Supplemental Figure 8). We don't think that accelerated flowering effects of our *FT2 β* overexpression plants are small, because the flowering time of wild-type plants (Bd21) is only around 47 days, while those of two types of transgenic plants are 59 and 53 days, respectively, which is significantly later than the wild-type by students t-test. Although we cannot exclude other mechanisms may be involved in the prevention of early flowering, it is reliable that *FT2 α* promotes while *FT2 β* prevents flowering from our molecular and phenotypic results using *FT2 α* and *FT2 β* overexpression and silencing transgenic plants (Figure 1, Figure 6, and Supplemental Figure 8).

(From the referee) There are new published data that shows that *FT2* is expressed in the apex and that therefore it may not have to move through the phloem to produce an effect. The authors need to show the level and time of expression of the two *FT2* variants in the apex.

Response: We are sorry that we cannot find a literature that shows *FT2* is expressed in the apex. Nevertheless, we did the experiments as the reviewer requested. We isolated *Brachypodium* meristems, examined the expression levels of the two *FT2* variants and found that both of *FT2* variants can be expressed in plant apex (Figure 2A below), even though the expression of *FT2* was very low when plants were grown 2 weeks (Figure 2A below). Intriguingly, we found that the expression ratio of *FT2 α* / *FT2 β* in apex is altered when plants grow from 4 weeks to 6 weeks (Figure 2B and 2C below), which is similar as it occurred in plant leaves, further suggesting that AS of *FT2* in *Brachypodium* is controlled by plant development.

Since whether *FT* gene is expressed in the apex is controversial and this part is not tightly related to our thesis, we omitted it in the main text because of the

space limitations. If the reviewer still requests to do so, we can add it in the next revised manuscript.

Figure 2 *FT2* AS is developmentally regulated in plant apex in *Brachypodium*

(A) RT-PCR analysis of *FT2α* and *FT2β* in 2-6 weeks plant apex.

(B) Relative expression of *FT2α* and *FT2β* in 2-6 weeks plant apex.

(C) Expression ratio of *FT2α* and *FT2β* in 2-6 weeks plant apex.

(From the referee) The authors used an artificial miRNA to specifically silence the *FT2β* variant. The idea is nice, and the result seems to agree with their model. However I think that they need extra evidence, as artificial miRNA could be unspecific. Also, there could be a problem with the design of the artificial miRNA. Almost always miRNAs start with a U at the 5' and this is very important for the miRNA loading in the proper silencing complex. Their artificial miRNA starts with an A, and this could send the miRNA to a different silencing complex. I know that they used the miR172 backbone, and miR172 is a particular miRNA that starts with A. However, the miR172 loading depends also in the secondary structure of the miRNA, and the authors need to explain clearly this potential pitfall. The effect on flowering is small (8 d) relative to the standard deviations presented in Figure 6F. The differences seem marginally significant (*). It would be better to increase the sample size to show a more conclusive effect on flowering

Response: Yes, we considered this case when we designed the experiments. Because there is no A nucleotide available for the design of artificial miRNA with a 5' U across the *FT2β* transcripts skipped the exon nucleotides from *FT2α* (Supplemental Figure 7A), we had to choose a miRNA with a 5' A to target *FT2β*. Because miR172 has been demonstrated to be efficiently loaded into AGO1 silencing complex, we used *MIR172a* as a backbone to generate artificial miRNA (Supplemental Figure 7B). We added more detailed

information and a supplemental figure in the revised manuscript to explain this clearly as the reviewer suggested.

To exclude the possibility that early flowering is not the result of unspecific effects by artificial miRFT2 β , we performed two aspects of additional experiments. Firstly, we introduced an amiRNA that was targeting FT2 α in *Brachypodium*, and found significantly delayed flowering phenotypes appeared in strong amiRFT2 α lines, suggesting FT2 β indeed play repressive roles in flowering (Supplemental Figure 8). Secondly, we predicted amiRFT2 β potential targets by bioinformatics analysis, of which pipeline is as similar as described before⁴. In addition to FT2 β , We predicted other three amiRFT2 β potential targets, including *Bradi4g05030*, *Bradi3g51880* and *Bradi2g19620*. The target sites of these three genes are not well matched with amiRFT2 β (all scores are 4) and their gene function annotations are not related to plant flowering (Figure 3a below). Additionally, we performed qPCR to examine their expressions and found all of their expressions are not changed in *amiRFT2 β* transgenic plants compared with wild-type plants (Figure 3b below). These results suggest that the early flowering of *amiRFT2 β* is not due to the effects of unspecific targets.

A

amiRFT2 β sequence: ACCUCGACGCCGACGAAC

No Annotation Score: 4.0 Bradi4g05027.1 | Symbols: CSLD5 | cellulose synthase-like D5 | Best Arabidopsis Hit: AT1G02730.1

Query: 1 ACCUCGACGCCGCGCGACGAAC 21
Sbjct: 223 GGGAGCUGCGCGG-GUCCAUG 204

No Annotation Score: 4.0 Bradi3g51880.1 | Symbols: | P-loop containing nucleoside triphosphate hydrolases superfamily protein | Best Arabidopsis Hit: AT3G45070.1

Query: 1 ACCUCGACGCCGCGCGACGAAC 21
Sbjct: 448 UGGAGCUGCGCGG-GUGCCUG 429

No Annotation Score: 4.0 Bradi2g19620.1 | Symbols: | Protein of unknown function, DUF584 | Best Arabidopsis Hit: AT4G21970.1

Query: 1 ACCUCGACGCCGCGCGACGAAC 21
Sbjct: 145 CGGAGCUGCGCGCGGGCGCG 125

B

Figure 3 Early flowering of *amiRFT2 β* transgenic plants is not result of its unspecific potential targets

(A) Sequences alignment of amiRFT2 β and its potential targets.

(B) qRT-PCR analysis of amiRFT2 β potential target gene expressions in wild-type and *amiRFT2 β* transgenic plants

We don't think the effects of *amiRFT2 β* is marginal, because the flowering time in *amiRFT2 β* plants are significantly earlier than wild-type Bd21-3 plants by students t-test. We used the T₃ homozygous transgenic lines with strong and weak *amiRFT2 β* to record their flowering time, and found the extents of early flowering phenotypes are correlated with the accumulation levels of mature artificial miRNA as well as the reduced *FT2 β* expressions (Figure 6). Furthermore, severe and mild *amiRFT2 α* transgenic plants didn't exhibit any early flowering phenotype (Supplemental Figure 8). Together, these results strongly demonstrate that the two splicing variants of *FT2* play opposite roles in flowering control.

(From the referee) Interactions: the authors show that FT2 β does not bind to FDL2 and GF14b and GF14c (FT2 α does interact). In Y3H FT2 β reduces the strength of the interaction between FT2 α and FDL2 and FT2 α and GF14. It is not clear to me how FT2 β interferes with the formation of the FAC if it does not bind to the GF14b and GF14c proteins that may mediate the interaction with FT2 α ... The authors indicate that this may be the result of direct interactions between FT2 β and FT2 α and FT1. The authors need to eliminate the possibility of indirect interactions mediated by the yeast 14-3-3 or the Nicotiana 14-3-3.

Response: From our yeast two-hybrid, BiFC and Co-IP experiments, we showed that FT2 α can interact with FD and 14-3-3s, while FT2 β can neither associate with FD nor 14-3-3s, suggesting that FT2 α and FT2 β may have different action molecular mechanisms. Therefore, we speculated that FT2 β may interfere the binding of florigen to FD and 14-3-3s. To test this possibility, we performed yeast three-hybrid and BiFC competition experiments, and indeed observed that FT2 β attenuated the binding capacity of FT2 α and FT1 to FD and 14-3-3s, supporting our hypothesis. We would like to point out that this repressive effect of FT2 β is not mediated by the yeast 14-3-3 or the Nicotiana 14-3-3, since FT2 β interferes not only the complexes formed by florigen and FD, but also those by florigen and GF14b and GF14c, the latter of which is not dependent on the scaffold proteins. Furthermore, we found that FT2 β can catch hold of FT2 α and FT1 to form heterodimers (Figure 4), directly resulting in reductive number FT2 α and FT1 proteins available to interact with FD and 14-3-3s to form FAC complexes.

(From the referee) Test binding of FT2 β to yeast 14-3-3. It is possible that FT2 β binds to any of the other multiple 14-3-3 proteins in Brachypodium and by the FAC complexes using these 14-3-3? Please expand the set of Brachypodium 14-3-3 tested with the two alternative splice forms of FT2.

Response: As the review suggested, we carried out additional yeast two-hybrid experiments, and tested the interactions of FT2 β to other four 14-3-3 proteins which belongs to different clades. We did not find any interactions occurred (Supplemental Figure 4B), further validating that FT2 β

cannot bind 14-3-3s. We added these results in the supplemental Figure 4B.

Reviewer #3

This review has some concerns about the interpretations of our qRT-PCR experiments and some minor points. We have addressed them in detail below.

(From the referee) The manuscript by Qin et al describes the characterization of a splice variant of a FLOWERING LOCUS T homologue in *Brachypodium distachyon*. The authors present two independent datasets on protein interaction that confirm that this splice variant, which they call FT2beta, encodes for a 5' internal deletion in the corresponding protein that makes it incapable of interacting with the B. distachyon proteins corresponding to previously characterized binding partners 14-3-3 and FD from *Arabidopsis thaliana*. In contrast, FT2alpha (the full-length version) and a second paralog can interact with both partners. Interestingly, FT2beta is not compromised in its capability to form heterodimers with FT2alpha or FT1. Furthermore, the authors show data of plants overexpressing ubiquitously either FT2beta or an artificial microRNA directed against FT2beta. These data show a clear and opposite effect on heading date for both scenarios. The authors propose as model that the truncated FT2beta version may act as competitive inhibitor to FT1 and FT2alpha, which act as florigen.

The authors also interrogate expression of both splice variants during development and observe that the ratios between FT2 beta and alpha change with age so that the more active FT2alpha predominates in older plants. These data seem plausible but the authors make the common mistake of over interpreting real-time qRT PCR data that are obtained with a relative quantification method. Therefore, some formulations have to be tuned down or, preferably, absolute quantifications of both transcripts need to be carried out (see detail below as major point).

Response: We have recalculated the relative quantification of our all qRT-PCR results as the reviewer suggested. Please see the revised figures and responses to the major points 1.

(From the referee) Interestingly, the alternative splicing of FT2 seems conserved among temperate grasses. It is important to stress in the discussion that this means that they are potentially regulating but that first, this needs to be shown in the other species as well.

The discussion is rather obvious and does not open up towards as yet unsolved future questions, e.g. in which tissue the competitive mechanisms is mainly located.

Response: We did more experiments and showed that *FT2 AS* is present in wheat, barley and *A. tauschii*, but absent in *Arabidopsis*, rice and maize, thus we propose that this mode of posttranscriptional regulation of FT is universal in temperate grasses. Please see our response to reviewer 1's second concern. Yes, we agree with the reviewer that our previous discussion section is limited. We have modified and added more contents in our revised manuscripts to provide more perspective as the reviewer suggested. Please see the revised discussion section.

(From the referee) Major points: 1. Page 7, second paragraph, third paragraph. Also page 8, third paragraph. All claims of „Gene A is expressed more than Gene B" are not supported by the method that seems to have been employed for quantification (not entirely detailed in methods). If no absolute quantification is used, qRT-PCR is a relative measure thus one control sample per primer pair is arbitrarily scaled to "1". Alternatively, the values are scaled to the average, but it is still an arbitrary scale. It is possible to say "gene A is more expressed in sample x as compared to a control", but not "A more/less than B". This also concerns Figure 5A, where the reference point is not obvious for FT2alpha and FT2beta. In Figure 5B, the ratio changes, that is true, but the „1" line is randomly set. It is still possible that FT2beta is expressed at negligible levels throughout development. Figure 5C and D have the same problem as A. The authors seem to be aware of how to calculate qRT values, because Figure 6C,D and G show a control (WT) sample as clear reference point (set to „1" in this case).

Response: Thank you for the review for pointing this out. Actually, we calculated the expression levels of *FT2 α* and *FT2 β* based on their comparison with the expression of *UBC18*, an endogenous control gene tested in parallel in qRT-PCR experiments. We agreed that qRT-PCR should be a relative measure, and thus recalculated all relative qRT-PCR values by setting the first sample "1" line in the revised manuscript as the reviewer suggested. Please see the revised figure 5 and the figure legends.

(From the referee) 2. All protein interaction assays involving the truncated „negative“ FT2beta protein. Theoretically, absence of interaction can be explained by an unstable protein (in both assays). Unfortunately in the case of the split YFP, the amount of protein that accumulates is a major source of artefacts and negative controls should be expressed at similar levels and in the same compartments to make that assay at least a little reliable. Protein stability could be visualized by showing comparative expression levels for plan YFP fusion of all proteins under study. Alternatively, the pull-down input, which is basically produced in the same expression system, could be shown as control across one western (now it looks as if FT2alpha accumulated more than FT2beta but that is probably due to the exposure time at different blots). It is true that the positive interaction detected between the truncated FT and the full-length FTs indicates that this protein does indeed accumulate to adequate levels, making at least a false negative unlikely.

Response: From our Co-IP experiments, we can see that both FT2 α and FT2 β are stable proteins (Figure 2C, Figure 4A below). We agree with the reviewer statement that the expression levels of FT2 α and FT2 β in Co-IP experiments looks like a little different is due to the different exposure time. To further validate this, we performed qRT-PCR and western blot analysis *FT2 α* and *FT2 β* expression and found *FT2 α* and *FT2 β* indeed are expressed the same level in the inoculated *N.benthamiana* leaves with same concentration of *A. tumefaciens* strains (Figure 4B below).

During the YFP-split experiment, we adjusted the same OD of *A. tumefaciens* containing FT2 β and the control GUS protein when we inoculated them into the *N. benthamiana* leaves. Also, we got the similar results from the experiments that were repeated three times, so the results of competition BiFC experiments are reliable. We totally agree with the reviewer that “the positive interaction detected between the truncated FT and the full-length FTs indicates that this protein does indeed accumulate to adequate levels, making at least a false negative unlikely”.

Figure 4 Same amounts of FT2 α and FT2 β are used in the protein interaction assays

(A) The full scan picture of Co-IP results from FT2 α and FT2 β with FDL2.

(B) qRT-PCR analysis of FT2 α and FT2 β in the inoculated *N.benthamiana* leaves with same OD of *A. tumefaciens* strains.

(From the referee) 3. Figure 4B. For the reasons detailed above, the „empty vector” control is useless as the fusion protein does not accumulate at levels comparable to fusion proteins.

Response: Instead of empty vector as a control as the reviewer suggested, we used GUS protein as a control and performed the experiments again. We still got the similar results. Please see the revised Figure 4B.

(From the referee) **Minor points 1.** The English needs to be fixed throughout the manuscript. Smaller changes, but too much for a peer review.

Response: We have improved our English carefully in the revised manuscript.

(From the referee) 2. Abstract. FT1 and FT2 are not orthologs of FT. They are paralogs and homologs of FT. Ortholog should be reserved for genes with conserved synteny.

Response: Thank you for pointing this out. We have changed the statement in

the revised abstract.

(From the referee) 3. Abstract. Revise text for precision. E.g. splice variants do not interact, the encoded proteins do. Last sentence: it is not a regulation machinery but a mode of regulation that was discovered; the data do not apply to "FT in plants" but to FT2 in *B. distachyon* and possibly other temperate grasses (last sentence).

Response: We have changed the statement in the revised abstract as the reviewer suggested.

(From the referee) 4. Citation 14 makes no sense in the context of FT cis regulation

Response: We have deleted this literature and added the correct one in the revised manuscript.

(From the referee) 5. The authors cite many reviews from the same groups (recent and older), which may be a little redundant. In contrast several key papers on FT regulation are missing.

Response: As the reviewer suggested, we have deleted the redundant review references and added some original key papers on *FT* regulation in the revised manuscript.

(From the referee) 6. AP2-like proteins do not bind in the 3'untranslated regions, but SMZ appears to bind downstream of FT based on the cited manuscripts.

Response: Sorry for the confusion. We have corrected the information about this in the revised manuscript.

(From the referee) 7. *Arabidopsis* in italics would mean the entire genus. If *Arabidopsis thaliana* aka *Arabidopsis* is meant, do not use italics but introduce as abbreviated term.

Response: We have changed *Arabidopsis* into *A. thaliana* in the revised manuscript to avoid confusion as the reviewer suggested.

(From the referee) 8. Not clear to me why the renaming of FTL1 and FT2 and of FTL2 as FT1 is less confusing because of VRN3 in wheat and barley. It seems to yet add another layer of confusion to rename these genes again (page 3).

Response: Sorry for the confusion. Since there are more identical amino acids of FTL2 than FTL1 in *B. distachyon* with florigen VRN3 in wheat and barley, we re-designated FTL1 as FT2 and FTL2 as FT1 based on the previous literatures. This will be clearer for readers to compare the proteins responsible for florigen activity in *Brachypodium*, wheat, barley and other temperate grasses. We have rewritten this in the new text and added an additional literature to explain it.

(From the referee) 9. Page 6 „using transient GUS protein as control" is confusing. Better „As control, GUS instead of FT2beta was expressed.. „ or something similar

Response: We have changed this as the reviewer suggested.

(From the referee) 10. Figure 1C. It would have been nice to see experimental data confirming that both splice variants use the same transcription start site and therefore also the same ATG (5'race). Since this has not been shown, indicate that in the legend - putative TSS for both transcripts.

Response: We added the TSS sites for both two *FT2* transcripts in the revised in Figure 1C. Please see the revised Figure 1C.

(From the referee) 11. Methods: RNA expression analysis: not clear how the analysis was done from this description (deltadelta CT? What was the reference point? The data show SD of n=3?)

Response: Yes, the reference indicates the SD of three repeats. We interpret this more clearly in the revised figure legends. We also wrote the methods in more detail regarding qRT-PCR in the revised methods section.

(From the referee) 12. I do not find the „Bright field" and „merged" pictures for Figure 2B,E and 3C and 4B all that informative.

Response: We showed the Bright field and merged pictures in these figures in order to clearly describe the subcellular fraction where the indicated proteins interact. We can delete them if the reviewer requests to do so.

Reference

- 1 Thomas, J. *et al.* Identification of an intronic splicing regulatory element involved in auto-regulation of alternative splicing of SCL33 pre-mRNA. *The Plant journal : for cell and*

molecular biology **72**, 935-946, doi:10.1111/tpj.12004 (2012).

- 2 Mandadi, K. K., Pyle, J. D. & Scholthof, K. B. Characterization of SCL33 splicing patterns during diverse virus infections in *Brachypodium distachyon*. *Plant signaling & behavior* **10**, e1042641, doi:10.1080/15592324.2015.1042641 (2015).
- 3 Mandadi, K. K. & Scholthof, K. B. Genome-wide analysis of alternative splicing landscapes modulated during plant-virus interactions in *Brachypodium distachyon*. *The Plant cell* **27**, 71-85, doi:10.1105/tpc.114.133991 (2015).
- 4 Wu, L. *et al.* Rice MicroRNA effector complexes and targets. *The Plant cell* **21**, 3421-3435, doi:10.1105/tpc.109.070938 (2009).

Reviewers' comments:

Reviewer #3 (Remarks to the Author):

For this revised version of Qin et al., I have mainly evaluated if the authors have addressed my previous concerns, while realizing that the other reviewers had more reservations about the quality of the presented data.

The reviewers addressed my technical concern only partially. Although Figure 1 is now improved and shows more proper representation of qRT-PCR data, the relevant Figure 5 is not and neither is Figure S2. The authors still make the mistake of directly comparing "levels" between the genes, while they can only make statements about fold-change between conditions per gene. Comparing to UBC18 only levels out differences in sample quality but does not provide a "scale". This criticism relates to the one made by reviewer 2, who doubts that the level of expression of the splice variant is relevant.

It would be very easy to prepare an absolute standard by cloning the fragments together in a 1/1 ratio by PCR or combine them in a plasmid. It is really not too much to ask it would allow to make the statement on relative expression levels of the variants.

My other technical concern was addressed by showing the protein levels of interaction partners in IP. Curiously, again, the authors use relative qRT-PCR to show that FT alpha and beta are equally expressed. Same problem - a statement that cannot be made with the assay used. However, as the relevant part is the western blot, the qRT result could just be deleted, it is not required.

Besides the methodological problems, there is still a need of language editing as stated before.

Reviewer #4 (Remarks to the Author):

The study by Wu et al. demonstrates that two different FT2 splice variants act as flowering activator and repressor in *Brachypodium*, respectively. While it is well known that FT-like genes may induce or delay floral development (shown for *Arabidopsis*, poplar, sugar beet etc.), the authors made the novel finding that both, activator and repressor could be coded for by the same gene. This mechanism of differential splicing of FT2 is specific for *Brachypodium*, barley and wheat and not found in more distant grasses or outside the grasses. I share the concerns of reviewer 2 about concluding from the FT-OE on the function of the two splice variants. However, I agree with the authors that the results of the RNAi experiments support the differential roles of the two FT splice variants. In addition, difference in the ratios of FT2 splice variants during development, binding assays with 14-3-3 and FD proteins and the analysis FT2 splice variants in other species make a very comprehensive study on the possible role of alternative FT2 splicing. The work added by the authors on the possible control of FT2 AS upon request of reviewer 1 is not very comprehensive and only mentioned in the discussion. In my opinion that should be rather the beginning of a new study.

I just have a few minor comments.

1. Introduction:

The authors describe the flowering pathway in *Arabidopsis thaliana*, but always write In plants....Please describe correctly which genes and pathways have been described for which plant.

2. l. 156 the authors write: Intriguingly, we found that all FT2 β overexpressing plants exhibited significantly later heading date compared with wild-type plants (Figure 1E). Since FT2 β -OE#1 and FT2 β -OE#3 exhibited the highest and lowest level of FT2 β overexpression, we chose them to do further analysis. This suggested that more transgenic plants were analysed. I could not find the flowering data of the other OE plants, only #1 and #2. In addition, the flowering and expression data is compared to the wild type rather than the null segregant.

3. l. 160 bolted 6 days later than....

4. What are the plant stages at the different weeks, when is floral transition? What is the juvenile and adult phase in *Brachypodium* (Fig. 7) and how does it relate to the weeks?

5. Is the model in Figure 7 in the leaf or is this supposed to take place in the shoot apical meristem. I suppose *Vrn1* expression was measured in the leaf. Please, specify the model.

6. In line 395 the authors add to the discussion some information on potential mechanisms for developmental differences in FT2 splicing. They write: growth stage-dependent transcriptional and posttranscriptional regulation of *SCL33* has occurred, which may directly or indirectly result in developmental regulation of FT2 AS. However, the data provided is not very comprehensive and does not provide enough data to elucidate the reasons for development dependent splicing of FT2.

7. L 380 the authors write: Nevertheless, although we detected that the ratio of FT2 β /FT2 α was influenced by ambient temperature changes, AS of FT2 may not have significant effects to temperature-mediated flowering control in *B. distachyon*, since increased ambient temperature is not a significant floral inductive signal in temperate grasses. I would be very careful with this argument, ambient temperature has a large effect on flowering in temperate grasses (even though it does not substitute for long photoperiods as in *Arabidopsis*). It is strange that the authors do not provide flowering data under lower and higher ambient temperatures themselves.

8. It is not so felicitous that all qRT-PCR data presents variation only for technical, but not for biological replicates.

Reviewer #5 (Remarks to the Author):

In this manuscript, the authors show that the alternative splicing of FT2 in *Brachypodium distachyon* results in two isoforms, FT2 α and FT2 β ; FT2 α is a positive regulator flowering while FT2 β delays the transition to reproductive development. That FT2 β is a negative regulator of flowering is supported by the overexpression of FT2 β which results in delayed flowering and its downregulation by an amiRNA which accelerates the transition to

flowering. The model proposed here, supported by yeast-two-hybrid, yeast-three-hybrid, and BIFC, is that FT2 β acts a negative regulator of flowering by interacting with FT1 and FT2 α . This interaction would thus impair the formation of the flowering activating complex, composed of FT, FD, and a 14-3-3 protein. The authors suggest that a high FT2 β /FT2 α ratio prevents flowering in young plants, and that the age-regulated increase in the FT2 α /FT2 β ratio would favor the transition to reproductive development in older plants.

Although most methodological approaches are suited to explore the biological question, some flaws might lead to an over-interpretation of the age-related modification of the ratio between FT2 α and FT2 β (RT-qPCR - see major point 1).

Major points

1) In our opinion, the demonstration of the switch in the balance between FT2 β and FT2 α is not carried out using proper methods.

A previous remark from reviewers said:

"The authors also interrogate expression of both splice variants during development and observe that the ratios between FT2 beta and alpha change with age so that the more active FT2alpha predominates in older plants. These data seem plausible but the authors make the common mistake of over interpreting real-time qRT PCR data that are obtained with a relative quantification method. Therefore, some formulations have to be tuned down or, preferably, absolute quantifications of both transcripts need to be carried out (see detail below as major point)."

The remark was likely misunderstood by the authors. Common RT-qPCR methods are used to determine the relative expression of a specific amplicon in different samples and assess whether the expression of this specific amplicon varies between the studied samples. The amplification of a specific target in a sample results in a Cq value, which is the number of cycles at which the fluorescence goes above a specific threshold. The fluorescence is the consequence of the incorporation of SYBR into the amplicon and varies according to the amplicon length and the AT content. Hence, different amplicons are likely to reach the established threshold at different cycles, even if their original abundances are identical. In addition, the efficiency of the primer pairs used for the amplification of the two isoforms (same REV but different FOR) are likely not 100% identical, and differences in efficiency might result in large differences in the Cq values. Together, these variations render unreliable

the comparison of the expression level of different amplicons through classical methods. Therefore, one cannot compare the level of two amplicons in the same graph (e.g. Figure 5A, 5C, 5D, S2). The FT2 β /FT2 α ratio (Fig. 5b) might also be affected by the reasons presented above, especially if there are variations in primer efficiencies.

Hence, it is not accurate to compare the expression level of these isoforms using relative

quantification by RT-qPCR. The authors must use a suitable quantification method if they want to compare the actual expression levels of isoforms (e.g. absolute quantification by qPCR, digital PCR, etc.). This methodological flaw must be corrected prior publication, or the conclusions regarding the age-related modification of the FT2 β /FT2 α needs to be revised. Figure 5 and S2 must be modified. More detailed information regarding how to perform absolute quantification of splice variants by RT-qPCR are readily available online.

MINOR POINTS

- 1) Please indicate in the manuscript the methodology used for the design of the amiRNAs, the prediction of off-targets genes, and add the data regarding the verification of the absence of regulation of these genes as supplementary information. Indeed, it is of outmost importance for the reader to have the confirmation early-flowering phenotypes are not due to off-target effects.
- 2) L21: FLOWERING LOCUS D (AT3G10390 in Arabidopsis) is not FD (AT4G35900). FD has no "full name". Furthermore, FLOWERING LOCUS D refers to a different gene! -see for example Science. 2003; 302(5651):1751-4. Regulation of flowering time by histone acetylation in Arabidopsis. "Here, we report that FLOWERING LOCUS D (FLD), one of six genes in the autonomous pathway, encodes a plant homolog of a protein found in histone deacetylase complexes in mammals." Other authors in flowering have made this FD = FLD mistake as well in actual publications.
- 3) The manuscript would benefit from some English editing.
- 4) The readers would benefit from data comparing the phenotype of FT2 β and FT2 α overexpressing lines in similar conditions (not only a reference to previous publications).
- 5) The authors state that FT1 and FT2 are able to form homo- and heterodimers. However, if the methods presented in the manuscript suggest an interaction, none of these data can discriminate between a dimer or a multimeric complex of a higher order. Please, indicate the size at which the bands of Figure 2 are observed.
- 6) In Material and Methods, the methodology used for the competitive BIFC assay is not clear: the authors state that FT2 β or GUS proteins were infiltrated together with the other constructs. I assume that what was infiltrated were plasmids, as suggested in the results section. Please clarify in the manuscript.
- 7) It would be informative to add the age at which tissues were harvested for RT-qPCR in the legend of the figures (e.g Figure 1g, Figure 6g).
- 8) For the RT-qPCR figures, it is still not clear whether the results are from one representative experiment (in which case the SD is based on technical replicates) or three biological replicates (in which case the SD is based on the means of the three biological replicates).
- 9) L172-L174. How the expression pattern of FT2 suggests that both isoforms of FT2 can interact with FD is unclear to me? Please revise.
- 10) In the discussion regarding FLM, consider discussing about the recent publication suggesting that the diminution of FLM activity is also regulated through nonsense-mRNA decay (Sureshkumar et al., 2016, Nature Plants).
- 11) Please indicate in figure legends the origin of the tissue from which RNAs were extracted: whole aerial part? All leaves? Leaf 3?
- 12) L437 - Do the authors can provide a reference indicating that a FT dimer would be too

large to move through vascular tissue? For instance, in Arabidopsis, the exclusion size of plasmodesmata between companion cell and sieve elements is >67kDa (Stadler, 2005), which more than twice the size of AtFT.

13) Indicating the number of plants used for the characterization of flowering under bar plot would help the reader make a decision about the statistical significance of the results.

14) It would be important to the reader to know the total number of independent transgenic lines you characterized (both amiRNA and OE lines).

Responses to Reviewers' Comments

We thank all reviewers for their constructive comments to improve our manuscript during second-round review process. We have performed additional experiments and revised the manuscript as reviewers suggested. We think we have addressed all the concerns made by the reviewers and hope this edition of manuscript would be published by the journal. The point-by point responses to the reviewers' comments are listed in detail below:

Reviewer #3

This reviewer still has some concerns about our analysis method for qPCR results. We have performed absolute standard curves by cloning the *FT2 α* and *FT2 β* fragments as the reviewer suggested, and did absolute quantification as well as recalculated of *FT2 α* and *FT2 β* expressions in the revised manuscript. This will do help readers know more clearly about *FT2 α* and *FT2 β* expression patterns.

(From the referee) For this revised version of Qin et al., I have mainly evaluated if the authors have addressed my previous concerns, while realizing that the other reviewers had more reservations about the quality of the presented data.

The reviewers addressed my technical concern only partially. Although Figure 1 is now improved and shows more proper representation of qRT-PCR data, the relevant Figure 5 is not and neither is Figure S2. The authors still make the mistake of directly comparing "levels" between the genes, while they can only make statements about fold-change between conditions per gene. Comparing to UBC18 only levels out differences in sample quality but does not provide a "scale". This criticisms relates to the one made by reviewer 2, who doubts that the level of expression of the splice variant is relevant.

It would be very easy to prepare an absolute standard by cloning the fragments together in a 1/1 ratio by PCR or combine them in a plasmid. It is really not too much to ask it would allow to make the statement on relative expression levels of the variants.

Response:

As the reviewer suggested, we diluted the plasmid DNA containing *FT2 α* ,

FT2β and *UBC18*, and used them as templates to perform qPCR, thereby generating a standard curve for quantification of *FT2α*, *FT2β* and *UBC18* transcripts, respectively (Figure S3a, Figure 1 below). Based on this standard curve, we made absolute quantifications of *FT2α*, *FT2β* and *UBC18* in each indicated experiments, and recalculated the expression levels and the ratios of *FT2α* and *FT2β* according to their quantifications to the corresponding absolute value of *UBC18* in different temperature environments and with different age. According to the absolute quantification data of *FT2α* and *FT2β*, we found that the ratio of *FT2β*/*FT2α* approaching to 1 was occurred when plants were grown to 4 weeks rather than 5 weeks. Thus, we performed additional experiments that determined diurnal expressions of *FT2α* and *FT2β* in 24-h period in 4-weeks-old plants. Please see the revised Supplementary Figure 2, Supplementary Figure 8 and Figure 5.

Figure 1 Standard curve for quantification of *FT2α*, *FT2β* and *UBC18* transcripts.

A series of 10-fold dilutions of the plasmid DNA containing *FT2α*, *FT2β* and *UBC18* was used to draw the absolute standard curve. The regression line from the dilution curve was used to determine the concentration of *FT2α*, *FT2β* and *UBC18*. y-axis means the value of threshold cycle. x-axis means DNA concentration.

(From the referee) My other technical concern was addressed by showing the protein levels of interaction partners in IP. Curiously, again, the authors use relative qRT-PCR to show that FT alpha and beta are equally expressed. Same problem - a statement that cannot be made with

the assay used. However, as the relevant part is the western blot, the qRT result could just be deleted, it is not required.

Response: We agree with the reviewer that same problem is made during relative qRT-PCR to show that *FT2 α* and *FT2 β* are equally expressed in Co-IP experiments. Thus, we changed to use absolute quantification of *FT2 α* and *FT2 β* and confirmed that the same amounts of protein were used in the interaction analysis in IP. Because the reviewer suggested "the relevant part was the western blot and the qRT result could be deleted" , we omitted this in the revised manuscript because of space limitations.

(From the referee) Besides the methodological problems, there is still a need of language editing as stated before.

Response: We carefully checked the text again and edited the inaccurate language in the revised manuscript, including the methodological part.

Reviewer #4

This reviewer has only some minor comments and we addressed all of them as below.

(From the referee) The study by Wu et al. demonstrates that two different FT2 splice variants act as flowering activator and repressor in *Brachypodium*, respectively. While it is well known that FT-like genes may induce or delay floral development (shown for *Arabidopsis*, poplar, sugar beet etc.), the authors made the novel finding that both, activator and repressor could be coded for by the same gene. This mechanism of differential splicing of FT2 is specific for *Brachypodium*, barley and wheat and not found in more distant grasses or outside the grasses. I share the concerns of reviewer 2 about concluding from the FT-OE on the function of the two splice variants. However, I agree with the authors that the results of the RNAi experiments support the differential roles of the two FT splice variants. In addition, difference in the ratios of FT2 splice variants during development, binding assays with 14-3-3 and FD proteins and the analysis FT2 splice variants in other species make a very comprehensive study on the possible role of alternative FT2 splicing. The work added by the authors on the

possible control of FT2 AS upon request of reviewer 1 is not very comprehensive and only mentioned in the discussion. In my opinion that should be rather the beginning of a new study.

I just have a few minor comments.

1. Introduction: The authors describe the flowering pathway in *Arabidopsis thaliana*, but always write In plants....Please describe correctly which genes and pathways have been described for which plant.

Response: We appreciate the reviewer for pointing this out. We have indicated the exact plant name that we referred in the revised manuscript.

(From the referee) 2. l. 156 the authors write: Intriguingly, we found that all FT2 β overexpressing plants exhibited significantly later heading date compared with wild-type plants (Figure 1E). Since FT2 β -OE#1 and FT2 β -OE#3 exhibited the highest and lowest level of FT2 β overexpression, we chose them to do further analysis. This suggested that more transgenic plants were analysed. I could not find the flowering data of the other OE plants, only #1 and #2. In addition, the flowering and expression data is compared to the wild type rather than the null segregant.

Response: We had eight independent FT2 β positive overexpression transgenic plants and all of them delayed flowering compared with wild-type plants . We chose FT2 β -OE#1 and FT2 β -OE#3 for further molecular analysis because they have the high and mild levels of FT2 β . We also compared the flowering time between wild-type and the null segregants of transgenic plants, and found they have similar flowering dates. We added an additional supplemental figure 3 to describe these in the revised manuscript.

(From the referee) 3. l. 160 bolted 6 days later than....

Response: Sorry for the confused sentence. We have changed it to "FT2 β -OE#1 bolted 12 days (59d versus 47d) and FT2 β -OE#3 bolted 6 days (53d versus 47d) later than wild-type plants in LD conditions."

(From the referee) 4. What are the plant stages at the different weeks, when is floral transition? What is the juvenile and adult phase in *Brachypodium* (Fig. 7) and how does it relate to the weeks?

Response: In our LD condition, Bd21 is usually heading when the plants grow to around 7 weeks old. Phase transition usually occurs at 4-5 weeks. We described this in the revised Figure 5a legend.

(From the referee) 5. Is the model in Figure 7 in the leaf or is this supposed to take place in the shoot apical meristem. I suppose *Vrn1* expression was measured in the leaf. Please, specify the model.

Response: Yes, the model presented in Figure 7 takes place in the shoot apical meristem. We measured the *VRN1* expression using the whole plant materials, including leaf and shoot apex. We specified this in the revised Figure legends.

(From the referee) 6. In line 395 the authors add to the discussion some information on potential mechanisms for developmental differences in *FT2* splicing. They write: growth stage-dependent transcriptional and posttranscriptional regulation of *SCL33* has occurred, which may directly or indirectly result in developmental regulation of *FT2* AS. However, the data provided is not very comprehensive and does not provide enough data to elucidate the reasons for development dependent splicing of *FT2*.

Response: We agree with the reviewer that our proposal of *SCL33*-mediated *FT2* splicing regulatory mechanism is not comprehensive and "should be the beginning of a new study". Since the reviewer 1 asked us to provide some hint about development-regulated *FT2* splicing mechanism during first-round revision, we did some preliminary experiments, referred to a paper, and thus gave a hypothesis that *SCL33* possibly mediated the *FT2* splicing events. Because the paper space is so limited and the reasons for development dependent splicing of *FT2* would be an independent story, we deleted this part in the revised manuscript as this reviewer suggested. If the

reviewer still require this, we can recover this information.

(From the referee) 7. L 380 the authors write: Nevertheless, although we detected that the ratio of FT2 β /FT2 α was influenced by ambient temperature changes, AS of FT2 may not have significant effects to temperature-mediated flowering control in *B. distachyon*, since increased ambient temperature is not a significant floral inductive signal in temperate grasses. I would be very careful with this argument, ambient temperature has a large effect on flowering in temperate grasses (even though it does not substitute for long photoperiods as in *Arabidopsis*). It is strange that the authors do not provide flowering data under lower and higher ambient temperatures themselves.

Response: We changed " since increased ambient temperature is not a significant floral inductive signal in temperate grasses " to " since increased ambient temperature has been shown to have limited influences on inductive flowering signal induction in temperate grasses " for accuracy. We did not examine the flowering date of *Brachypodium* under different ambient temperatures because others have reported this before and we referred to their paper in our manuscript.

(From the referee) 8. It is not so felicitous that all qRT-PCR data presents variation only for technical, but not for biological replicates.

Response: Thanks for the reviewer reminding of this. Actually, we performed all qRT-PCR analysis at least three biological replicates with three technical repeats. We presented it in the method section. qRT-PCR data showed in the figure is one representative biological result with three technical replicates. We clearly indicated this in the revised figure legends as well as in the method section.

Reviewer #5

This reviewer has some concerns about our qRT-PCR data analysis and some minor points. We provide responses as below.

(From the referee) In this manuscript, the authors show that the alternative splicing of FT2 in *Brachypodium distachyon* results in two isoforms, FT2 α and FT2 β ; FT2 α is a positive regulator of flowering while FT2 β delays the transition to reproductive development. That FT2 β is a negative regulator of flowering is supported by the overexpression of FT2 β which results in delayed flowering and its downregulation by an amiRNA which accelerates the transition to flowering. The model proposed here, supported by yeast-two-hybrid, yeast-three-hybrid, and BIFC, is that FT2 β acts as a negative regulator of flowering by interacting with FT1 and FT2 α . This interaction would thus impair the formation of the flowering activating complex, composed of FT, FD, and a 14-3-3 protein. The authors suggest that a high FT2 β /FT2 α ratio prevents flowering in young plants, and that the age-regulated increase in the FT2 α /FT2 β ratio would favor the transition to reproductive development in older plants.

Although most methodological approaches are suited to explore the biological question, some flaws might lead to an over-interpretation of the age-related modification of the ratio between FT2 α and FT2 β (RT-qPCR - see major point 1).

Major points

1) In our opinion, the demonstration of the switch in the balance between FT2 β and FT2 α is not carried out using proper methods. A previous remark from reviewers said: "The authors also interrogate expression of both splice variants during development and observe that the ratios between FT2 beta and alpha change with age so that the more active FT2alpha predominates in older plants. These data seem plausible but the authors make the common mistake of over interpreting real-time qRT PCR data that are obtained with a relative quantification method. Therefore, some formulations have to be tuned down or, preferably, absolute quantifications of both transcripts need to be carried out (see detail below as major point)."

The remark was likely misunderstood by the authors. Common RT-qPCR methods are used to determine the relative expression of a specific amplicon in different samples and assess whether the expression of this specific amplicon varies between the studied samples. The amplification of a specific target in a sample results in a Cq value, which is the number of cycles at which the fluorescence goes above a specific threshold. The fluorescence is the consequence of the incorporation of SYBR into the amplicon and varies according to the

amplicon length and the AT content. Hence, different amplicons are likely to reach the established threshold at different cycles, even if their original abundances are identical. In addition, the efficiency of the primer pairs used for the amplification of the two isoforms (same REV but different FOR) are likely not 100% identical, and differences in efficiency might result in large differences in the C_q values. Together, these variations render unreliable the comparison of the expression level of different amplicons through classical methods. Therefore, one cannot compare the level of two amplicons in the same graph (e.g. Figure 5A, 5C, 5D, S2). The FT2 β /FT2 α ratio (Fig. 5b) might also be affected by the reasons presented above, especially if there are variations in primer efficiencies.

Hence, it is not accurate to compare the expression level of these isoforms using relative quantification by RT-qPCR. The authors must use a suitable quantification method if they want to compare the actual expression levels of isoforms (e.g. absolute quantification by qPCR, digital PCR, etc.). This methodological flaw must be corrected prior publication, or the conclusions regarding the age-related modification of the FT2 β /FT2 α needs to be revised. Figure 5 and S2 must be modified. More detailed information regarding how to perform absolute quantification of splice variants by RT-qPCR are readily available online.

Response: We made an absolute standard for the amplification of *FT2 α* and *FT2 β* , and recalculated all qRT-PCR data based on the absolute quantification method. We modified our Figure 5 and Figure S2 in the revised manuscript. Please see our responses to reviewer 3.

(From the referee) MINOR POINTS : 1) Please indicate in the manuscript the methodology used for the design of the amiRNAs, the prediction of off-targets genes, and add the data regarding the verification of the absence of regulation of these genes as supplementary information. Indeed, it is of outmost importance for the reader to have the confirmation early-flowering phenotypes are not due to off-target effects.

Response: We added an additional figure to describe our amiRNA design, and showed that the early-flowering phenotypes in *amiRFT2 β* are not due to the off-target effects. Please see the Supplemental Figure 10.

(From the referee) 2) L21: FLOWERING LOCUS D (AT3G10390 in Arabidopsis) is not FD (AT4G35900). FD has no "full name". Furthermore, FLOWERING LOCUS D refers to a different gene! -see for example Science. 2003; 302(5651):1751-4. Regulation of flowering time by histone acetylation in Arabidopsis. "Here, we report that FLOWERING LOCUS D (FLD), one of six genes in the autonomous pathway, encodes a plant homolog of a protein found in histone deacetylase complexes in mammals." Other authors in flowering have made this FD = FLD mistake as well in actual publications.

Response: Sorry about that. We delete the statement of FLOWERING LOCUS D in the revised manuscript.

3) The manuscript would benefit from some English editing.

Response: We carefully checked the language again and edited the errors and the confusions.

(From the referee) 4) The readers would benefit from data comparing the phenotype of FT2 β and FT2 α overexpressing lines in similar conditions (not only a reference to previous publications).

Response: The extremely early-flowering phenotype of FT2 α overexpressing transgenic plants has been clearly presented by us and others, so we referred to our and other groups' previous papers about this.

(From the referee) 5) The authors state that FT1 and FT2 are able to form homo- and heterodimers. However, if the methods presented in the manuscript suggest an interaction, none of these data can discriminate between a dimer or a multimeric complex of a higher order. Please, indicate the size at which the bands of Figure 2 are observed.

Response: As the reviewer suggested, we added a full scan picture of the western blot as supplemental figure 14 and marked molecular size of the proteins shown in Figure 2c. We get the conclusion that FT1 and FT2 are able to form homo- and heterodimers from the results of yeast two-hybrid assays (Figure 4 and S7). We are sorry that it is difficult to measure whether a protein

can form dimers from western blot, because the gel used for blot is SDS-PAGE and the protein is denatured.

(From the referee) 6) In Material and Methods, the methodology used for the competitive BIFC assay is not clear: the authors state that FT2 β or GUS proteins were infiltrated together with the other constructs. I assume that what was infiltrated were plasmids, as suggested in the results section. Please clarify in the manuscript.

Response: The statement has been changed to " For protein binding competition assay, the *A. tumefaciens* with FT2 β plasmid was co-infiltrated with equal volume of the indicated constructs. *A. tumefaciens* harboring *GUS* in place of FT2 β was infiltrated in parallel as a control."

(From the referee) 7) It would be informative to add the age at which tissues were harvested for RT-qPCR in the legend of the figures (e.g Figure 1g, Figure 6g).

Response: We indicated the age and the tissues of plants that were used in qPCR analysis in the revised figure legends.

(From the referee) 8) For the RT-qPCR figures, it is still not clear whether the results are from one representative experiment (in which case the SD is based on technical replicates) or three biological replicates (in which case the SD is based on the means of the three biological replicates).

Response: We did each qRT-PCR analysis at least three biological repeats with three technical replicates. Every biological replicates have similar results. Our qRT-PCR results showed in the figure is from one representative experiment and SD is based on three technical replicates. We clarified this in the revised figure legends.

(From the referee) 9) L172-L174. How the expression pattern of FT2 suggests that both isoforms of FT2 can interact with FD is unclear to me? Please revise.

Response: We deleted this sentence to avoid the confusion .

(From the referee) 10) In the discussion regarding FLM, consider discussing about the recent publication suggesting that the diminution of FLM activity is also regulated through nonsense-mRNA decay (Sureshkumar et al., 2016, Nature Plants).

Response: We added some information about this in the revised discussion section.

(From the referee) 11) Please indicate in figure legends the origin of the tissue from which RNAs were extracted: whole aerial part? All leaves? Leaf 3?

Response: We provided the information for the tissues that we used for RNA extraction in the revised figure legends.

(From the referee) 12) L437 - Do the authors can provide a reference indicating that a FT dimer would be too large to move through vascular tissue? For instance, in Arabidopsis, the exclusion size of plasmodesmata between companion cell and sieve elements is >67kDa (Stadler, 2005), which more than twice the size of AtFT.

Response: Sorry, our statement is not accurate, and we deleted this in the revised discussion section to avoid misunderstanding.

(From the referee) 13) Indicating the number of plants used for the characterization of flowering under bar plot would help the reader make a decision about the statistical significance of the results.

Response: We added the number of plants that were used for characterization of flowering in the characterization of flowering date in the figure legends.

(From the referee) 14) It would be important to the reader to know the total number of independent transgenic lines you characterized (both amiRNA and OE lines).

Response: We indicated the number of independent transgenic plants in the revised manuscript.

Reviewers' comments:

Reviewer #4 (Remarks to the Author):

The authors have introduced all changes requested by me in the earlier review. I have only one further comment. The significant changes in FT2 expression and in FT2a/b ratios occur between week 5 and 6 (Fig. 5). The authors write that floral transition occurs between week 4 and 5. They conclude from their data that: these results suggest that FT2 α and FT2 β may have different effects

on floral transition in *B. distachyon*. Since floral transition occurs before the changes in FT2a/b ratios are observed, FT2 is likely not so important for floral transition, but rather for further inflorescence development? Please, check if your conclusions are logical.

A language edit may be still useful, check the use of articles and prepositions.

It would be useful if the authors could provide the accession numbers for the genes in the supplementary Table 1.

Reviewer #5 (Remarks to the Author):

In this revised manuscript, the authors show that the alternative splicing of FT2 in *Brachypodium distachyon* results in two isoforms, FT2 α and FT2 β ; FT2 α is a positive regulator of flowering while FT2 β delays the transition to reproductive development. That FT2 β is a negative regulator of flowering is supported by the overexpression of FT2 β which results in delayed flowering and by its downregulation by an amiRNA accelerating the transition to flowering. The model proposed here, supported by yeast-two-hybrid, yeast-three-hybrid, and BIFC, is that FT2 β acts as a negative regulator of flowering by interacting with FT1 and FT2 α . This interaction would thus impair the formation of the flowering activating complex, composed of FT, FD, and a 14-3-3 protein. The authors suggest that a high FT2 β /FT2 α ratio prevents flowering in young plants, and that the age-regulated increase in the FT2 α /FT2 β ratio would favor the transition to reproductive development in older plants.

In our opinion, this revised version addressed most of the concerns raised by the reviewers. However, we still have some remarks; the most important one is regarding the absolute quantification of gene expression levels by RT-qPCR.

Major remark

1 - The RT-qPCR data seems more accurate in the revised manuscript. However, the readers need to be sure that the dilution curves used for the assessment of absolute gene expression levels were performed in every plate along the samples of interest. This is essential to avoid inter-plate variability (i.e. variability between two different runs), which would likely result in errors in the absolute quantification of gene expression levels. In turn, it would result in misleading data when comparing two different pairs of primers.

In short, for each experiment using absolute gene quantification, the RT-qPCR reaction plate

including samples of interest should have included a standard curve for the pair of primers used. If not, the comparison of the expression levels of one AS form versus the other might not be accurate.

The authors, if they did so, must indicate in the material and methods that they ran the standard curve along with the samples of interest for each pair of primers. If they did not run such standard curve on every plate, but only once on an independent plate, then the comparison of the levels of the two isoforms might not be accurate.

Minor remarks

1 - Although the language editing in the revised version of the manuscript corrected many mistakes, we think that additional editing is still necessary prior publication.

2 - The analyses of flowering time are only showed as the numbers of days to heading. However, many factors can influence the timing of flowering (germination time, endogenous development variability, etc.). Hence, the authors should either add information about the number of leaves on the main shoot at heading or clearly and specifically state in the Results that the plants displayed similar rates of vegetative development.

3 - In the figure presenting the absolute quantifications of RT-qPCR data (Figure S2B; Figure S8, Figure S16), the Y scale is likely incorrect, as the origin of the continuous scale should not be "0."

4 - In Figure 1F, 6F, and S3A: the authors should display the whole Y-scale, from 0 to 70, instead of starting it at 20/30 days. Such representation is a bit misleading for the reader.

5 - In figure S3A and S3B, the authors should use the same Y-scale, as it is confusing when comparing the two panels.

6 - Reviewer 4 mentioned that a clarification should be made regarding the use of the term "juvenile". The juvenile-to-adult and the vegetative-to-reproductive phase transitions are often uncoupled. To use the "juvenile" term, the authors should provide physiological/morphological criteria showing when the juvenile-to-adult phase transition occurs in *Brachypodium distachyon*. Alternatively, the authors can use another terminology.

7 - In Figure S11, it would be useful for the reader to see the data for all four independent lines (as provided in Fig. S3 for the overexpression lines).

8 - In several figure legends, the authors use "qPCR" instead of RT-qPCR (e.g. Fig. 6, Fig S11c).

9 - The authors do not reference the right Figure in the following paragraph:

"To exclude the possibility that the early flowering in amiRFT2 β is due to the unspecific

effects of artificial miRNA, we generated amiRFT2 α transgenic lines, which is specifically silencing FT2 α expressions by introducing another artificial miRNA targeting the intron region of FT2 α where is different from FT2 β (Supplementary Fig. 7)."

10 - The experimental replicates showed in Figures S14 to S18 should be referenced in the legends of the figures showing the first replicate (currently, they are not referenced anywhere).

11 - The following paragraph is misleading:

"To determine the role, if any, of different FT2 variants in control of flowering, we attempted to overproduce FT2 α and FT2 β in *B. distachyon*, respectively. As previously described by us and others, ectopic expression of intact FT2 (FT2 α) in *B. distachyon* leads to very early flowering and arrest vegetative growth, demonstrating the positive florigen activity of FT2 α _{14,40}. Next, we generated FT2 β overexpression (FT2 β -OE) transgenic plants and determined the flowering time in their T1 generations under LD conditions."

The reader wants to look at the FT2 α OE phenotype when reading this paragraph, and the data are not provided. Revise the sentence or indicate "data not shown."

12 - We also have a few remarks regarding the introduction and the discussion:

12A. In the last sentence of the introduction, the authors state:

"Together, these results reveal a novel molecular mode of FT post-transcriptional regulation in plants."

However, the manuscript establishes that such splicing was not observed in other model plants outside temperate grasses. In our opinion, the authors should rather conclude their introduction as they conclude the abstract, by:

"of FT post-transcriptional regulation in temperate grasses".

12B. The last sentence of the discussion states:

"In respect that multiple alternatively spliced forms of FT orthologous genes have been detected in *Platanus acerifolia*⁴⁸, it will be interesting to determine whether blocking flowering complexes formation by splicing variants is a universal mechanism of FT regulation in other plants."

However, the authors previously said that they did not observe such mechanisms in several other model plants. Hence, it cannot be universal. Please rephrase.

12C. For the interest of the authors, we would like to indicate that TFL1 – like BFT - also very likely represses FT activity through its interaction with FD (Hanano and Goto, 2011; Jaeger et al., 2013). In addition, TFL1 was shown to participate to the prevention of

flowering in immature meristems in *A. thaliana* (Matsoukas et al., 2013), *A. alpina* (Wang et al., 2011 - Plant Cell), and even in apple trees (Kotoda et al., 2006).

12D. In the discussion, the authors state:

"Moreover, the repressive effect of FT2 β is greater than BFT, since FT2 β disrupts not only FT2 α - but also FT1-mediated flowering initiation complex".

We are not sure of the relevance of this affirmation since BFT also likely prevents the activity of TSF, the homolog of FT, and that the comparison of the strength of a repressive signal in two model species that diverged 120 Million years ago seems difficult.

12E. The second paragraph of the discussion seems too long, as the conclusion is that the mechanisms mentioned above are very likely not relevant in *Brachypodium distachyon*.

Responses to Reviewers' Comments

We thank the reviewers for their constructive comments to improve our manuscript during third-round review process. We have performed additional experiments and revised the manuscript as the reviewer suggested. We have addressed all the concerns made by the reviewers and hope this edition of manuscript would be published by the journal. The point-by point responses to the reviewers' comments are listed in detail below:

Reviewer #4

(From the referee) The authors have introduced all changed requested by me in the earlier review. I have only one further comment. The significant changes in FT2 expression and in FT2a/b ratios occur between week 5 and 6 (Fig. 5). The authors write that floral transition occur between week 4 and 5. The conclude from their data that: these results suggest that FT2 α and FT2 β may have different effects on floral transition in *B. distachyon*. Since floral transition occurs before the changes in FT2a/b ratios are observed, FT2 is likely not so important for floral transition, but rather further inflorescence development? Please, check if your conclusions are logic.

Response: Plants will enter floral transition phase when their functional florigen accumulate to a threshold, so the floral transition of *Brachypodium* does not strictly occurs when the ratio of FT2 α and FT2 β changes, but takes place when the plants accumulate enough flowering initiation complexes. We propose that if there were no FT2 β generated to repress flowering initiation complexes formation when plants were in early vegetative phase, the plants would flower precociously, which is harmful to propagate their seeds. Moreover, we did not observe abnormal inflorescence development in gain- or loss- of FT2 β function transgenic plants. Therefore, we think that FT2 AS is involved in floral transition rather than in flower development.

(From the referee) A language editing may be still useful, check the use of articles and prepositions.

Response: Thank you for pointing out this and we have asked a language editor to help us edit English.

(From the referee) It would be useful if the authors could provide the accession numbers for the genes in the supplementary Table 1.

Response: We provided gene accession numbers behind the method section.

Reviewer #5

(From the referee) In this revised manuscript, the authors show that the alternative splicing of FT2 in *Brachypodium distachyon* results in two isoforms, FT2 α and FT2 β ; FT2 α is a positive regulator flowering while FT2 β delays the transition to reproductive development. That FT2 β is a negative regulator of flowering is supported by the overexpression of FT2 β which results in delayed flowering and by its downregulation by an amiRNA accelerating the transition to flowering. The model proposed here, supported by yeast-two-hybrid, yeast-three-hybrid, and BIFC, is that FT2 β acts a negative regulator of flowering by interacting with FT1 and FT2 α . This interaction would thus impair the formation of the flowering activating complex, composed of FT, FD, and a 14-3-3 protein. The authors suggest that a high FT2 β /FT2 α ratio prevents flowering in young plants, and that the age-regulated increase in the FT2 α /FT2 β ratio would favor the transition to reproductive development in older plants.

In our opinion, this revised version addressed most of the concerns raised by the reviewers. However, we still have some remarks; the most important one is regarding the absolute quantification of gene expression levels by RT-qPCR.

1 - The RT-qPCR data seems more accurate in the revised manuscript. However, the readers need to be sure that the dilution curves used for the assessment of absolute gene expression levels were performed in every plate along the samples of interest. This is essential to avoid inter-plate variability (i.e. variability between two different runs), which would likely result in errors in the absolute quantification of gene expression levels. In turn, it would result in misleading data when comparing two different pairs of primers.

In short, for each experiment using absolute gene quantification, the RT-qPCR reaction plate including samples of interest should have included a standard curve for the pair of primers used. If not, the comparison of the expression levels of one AS form versus the other might

not be accurate.

The authors, if they did so, must indicate in the material and methods that they ran the standard curve along with the samples of interest for each pair of primers. If they did not run such standard curve on every plate, but only once on an independent plate, then the comparison of the levels of the two isoforms might not be accurate.

Response: Actually, our absolute standard curve of each transcript was carried out based on three independent experiments from three different plates, and the results are very consistent (shown in Figure 1 below). Additionally, our standard curves for the indicated genes were made from the average value of three independent experiment results. We placed these information as Supplementary Figure S2 in the revised manuscript (Please also see in Figure 1 below).

Figure 1 Standard curves for quantification of *FT2α*, *FT2β* and *UBC18* transcripts.

- a) Ct values of qRT-PCR analyses using dilutions of *FT2α*, *FT2β* and *UBC18* template DNA in three independent experiments.
- b) Standard curves for quantification of *FT2α*, *FT2β* and *UBC18* transcripts, respectively. Each threshold cycle at y-axis is the average Ct value from three replicates shown in a.

The reviewer suggested us that we should perform absolute quantification based on a standard curve running on a same PCR plate. Unfortunately, this is technically impossible for our experiment which is to simultaneously compare *FT2α* and *FT2β* absolute expression in 1- to 7-week-old plants. Because we have 7 samples (1 to 7 weeks old plants), 3 genes (*FT2α*, *FT2β* and *UBC18*) and three replicates for each gene in an experiment, which means that at least 63 ($7 \times 3 \times 3$) wells are required on each plate. Meanwhile, we need 6 dilutions of plasmid with three repeats for each standard curve determination, which means that 54 ($6 \times 3 \times 3$) wells are also required on a plate. Therefore, at least 117 ($63 + 54$) wells are theoretically required in all for examination of *FT2α* and *FT2β* absolute expressions with standard curve on the same plate. Since there is only qPCR machine with 96-wells on one plate available to us, it is technically impossible to do such an experiment. Similar case takes place for the experiments of *FT2α* and *FT2β* diurnal expression determination shown in Figure 5C and 5D.

To confirm our comparison of the levels of *FT2α* and *FT2β* is accurate, we redesigned and selected the key samples (3- to 7-week-old plants) which can fit on a single plate to do the experiment. We performed such qRT-PCR as the method reviewer suggested, and added these results to revised Figure S12 (see Figure 2 below), and also provided the information that how we arranged the wells on the plate and raw data showing Ct values per well. These results verified that our qRT-PCR results are accurate.

Figure 2 Absolute quantification of *FT2α*, *FT2β* and *UBC18* expressions in three- to seven-week-old *B. distachyon*.

- a) The qRT-PCR reaction plate containing the samples of *B. distachyon* with different age and the indicated serial-dilution plasmids for making standard curve of the used primers. The wells underlined with red, blue and black color represent the wells for *FT2α*, *FT2β* and *UBC18*, respectively. STD, Standards; w, week samples.
- b) The raw data of Ct values of per well arranged on the plate shown in a).
- c) Standard curve for quantification of *FT2α*, *FT2β* and *UBC18* transcripts.
- d) Absolute value of *FT2α*, *FT2β* and *UBC18* in three- to seven-week-old *B. distachyon* plants.
- e) Expression levels of *FT2α* and *FT2β* in *B. distachyon* with the indicated age.
- f) *FT2β/FT2α* ratios in three- to seven-week-old *B. distachyon* plants.

Additionally, we applied qRT-PCR analysis according to the double standard curve method, and each qRT-PCR absolute result obtained is not only from the standard curve of each AS transcript, but also from its corresponding *UBC18* absolute value in each sample, which do good to avoid the variability. Also, our qRT-PCR results from three independent biological replicates with three technique repeats are consistent and solid (Fig S3 and Fig S11). Thus, our qRT-PCR data are reliable.

(From the referee) Minor remarks 1-Although the language editing in the revised version of the manuscript corrected many mistakes, we think that additional editing is still necessary prior publication.

Response: Thank you for pointing out this and we have asked a language editor to help us improve English.

(From the referee) 2 - The analyses of flowering time are only showed as the numbers of days to heading. However, many factors can influence the timing of flowering (germination time, endogenous development variability, etc.). Hence, the authors should either add information about the number of leaves on the main shoot at heading or clearly and specifically state in the Results that the plants displayed similar rates of vegetative development.

Response: We used the numbers of days from the day of emergence of plant coleoptile to the first day when emergence of the spike is detected to record the flowering time in temperate grasses including *Brachypodium*, which is also referred in several published papers. As the reviewer suggested, to diminish misleading, we clearly and specifically stated in the revised results section that the transgenic and wild-type plants displayed similar rates of vegetative development.

(From the referee) 3 – In the figure presenting the absolute quantifications of RT-qPCR data (Figure S2B; Figure S8, Figure S16), the Y scale is likely incorrect, as the origin of the continuous scale should not be “0.”

Response: We checked that the origins of Y-scale shown in these figures indeed should be "0".

(From the referee) 4 – In Figure 1F, 6F, and S3A: the authors should display the whole Y-scale, from 0 to 70, instead of starting it at 20/30 days. Such representation is a bit misleading for the reader.

Response: We revised all of them in the revised figures.

(From the referee) 5 - In figure S3A and S3B, the authors should use the same Y-scale, as it is confusing when comparing the two panels.

Response: We used the same Y-scale in these two revised figures.

(From the referee) 6 - Reviewer 4 mentioned that a clarification should be made regarding the use of the term "juvenile". The juvenile-to-adult and the vegetative-to-reproductive phase transitions are often uncoupled. To use the "juvenile" term, the authors should provide physiological/morphological criteria showing when the juvenile-to-adult phase transition occurs in *Brachypodium distachyon*. Alternatively, the authors can use another terminology.

Response: Sorry for the inaccuracy. We used the vegetative to reproductive term in the revised manuscript to avoid confusion.

(From the referee) 7 - In Figure S11, it would be useful for the reader to see the data for all four independent lines (as provided in Fig. S3 for the overexpression lines).

Response: We provided an additional figure (Fig S17) to present the flowering data for all four transgenic lines.

(From the referee) 8 – In several figure legends, the authors use "qPCR" instead of RT-qPCR (e.g. Fig. 6, Fig S11c).

Response: We used qRT-PCR in the whole revised manuscript.

(From the referee) 9 - The authors do not reference the right Figure in the following paragraph:

"To exclude the possibility that the early flowering in amiRFT2 β is due to the unspecific effects of artificial

miRNA, we generated amiRFT2 α transgenic lines, which is specifically silencing FT2 α expressions by introducing another artificial miRNA targeting the intron region of FT2 α where is different from FT2 β (Supplementary Fig. 7)."

Response: Sorry, the reference should be supplementary Fig. 9 before.

(From the referee) 10 - The experimental replicates showed in Figures S14 to S18 should be referenced in the legends of the figures showing the first replicate (currently, they are not referenced anywhere).

Response: As the reviewer suggested, we referenced them in the revised legend of the figures when showing the first replicate, and integrated the information about all three repeats in the same supplementary figure in the revised manuscript.

(From the referee) 11 - The following paragraph is misleading: "To determine the role, if any, of different FT2 variants in control of flowering, we attempted to overproduce FT2 α and FT2 β in *B. distachyon*, respectively. As previously described by us and others, ectopic expression of intact FT2 (FT2 α) in *B. distachyon* leads to very early flowering and arrest vegetative growth, demonstrating the positive florigen activity of FT2 α 14,40. Next, we generated FT2 β overexpression (FT2 β -OE) transgenic plants and determined the flowering time in their T1 generations under LD conditions."

The reader wants to look at the FT2 α OE phenotype when reading this paragraph, and the data are not provided. Revise the sentence or indicate "data not shown."

Response: Because the phenotypes of transgenic plants with ectopic expression of *FT2 α* were clearly stated by us and others in previous papers, here, we referred to them and indicated "data not shown" as the reviewer suggested.

(From the referee) 12 - We also have a few remarks regarding the introduction and the discussion:

12A. In the last sentence of the introduction, the authors state: "Together, these results reveal a novel molecular mode of FT post-transcriptional regulation in plants."

However, the manuscript establishes that such splicing was not observed in other model plants outside temperate grasses. In our opinion, the authors should rather conclude their introduction as they conclude

the abstract, by: "of FT post-transcriptional regulation in temperate grasses".

Response: We revised "in plants" to "in temperate grasses" as the reviewer suggested.

(From the referee) "In respect that multiple alternatively spliced forms of FT orthologous genes have been detected in *Platanus acerifolia*, it will be interesting to determine whether blocking flowering complexes formation by splicing variants is a universal mechanism of FT regulation in other plants."

However, the authors previously said that they did not observe such mechanisms in several other model plants. Hence, it cannot be universal. Please rephrase.

Response: We revised it to "In respect that multiple alternatively spliced forms of *FT* orthologous genes have been detected in *Platanus acerifolia*, it will be interesting to determine whether the blocking flowering complex formation by splice variants is the same *FT* regulatory mechanism in trees."

(From the referee) 12C. For the interest of the authors, we would like to indicate that TFL1 – like BFT - also very likely represses FT activity through its interaction with FD (Hanano and Goto, 2011; Jeager et al., 2013). In addition, TFL1 was shown to participate to the prevention of flowering in immature meristems in *A. thaliana* (Matsoukas et al., 2013), *A. alpina* (Wang et al., 2011 - Plant Cell), and even in apple trees (Kotoda et al., 2006).

Response: We added some information about TFL1 which also represses FT activity through the interaction with FD and referred to the paper that the reviewer suggested.

(From the referee) 12D. In the discussion, the authors state: "Moreover, the repressive effect of FT2 β is greater than BFT, since FT2 β disrupts not only FT2 α - but also FT1-mediated flowering initiation complex".

We are not sure of the relevance of this affirmation since BFT also likely prevents the activity of TSF, the homolog of FT, and that the comparison of the strength of a repressive signal in two model species that diverged 120 Million years ago seems difficult..

Response: We deleted this sentence in the revised manuscript to avoid confusion.

(From the referee) 12E. The second paragraph of the discussion seems too long, as the conclusion is that the mechanisms mentioned above are very likely not relevant in *Brachypodium distachyon*.

Response: We deleted some contents which are not related to *Brachypodium* flowering mechanism in this paragraph as the reviewer suggested.

REVIEWERS' COMMENTS:

Reviewer #5 (Remarks to the Author):

1 - L31/32 : This sentence is confusing, as one may interpret that the change in isoform also acts on flowering in other temperate grasses. Please precise that the mechanism you are referring to is the alternative splicing of FT2 β

2 - L209-211: The sentence is not clear: the fact that FT2 β does not bind to FD/14-3-3 does not imply that it represses flowering, but that it might act through another mechanism, for example by interfering with the formation of the flowering activating complex. Please revise.

3 - Figure 1 & Figure 6: please indicate the level of significance for * and ** in the legend.

4 - Note that colors used in Figure 6 (FT2 α vs. FT2 β) are impossible to discriminate for color blind people (or when printed in B&W). Maybe consider using more contrasting colors and/or use different symbols for the gene expression kinetics.

5 - Figure S14: the y-axis annotation is lacking on the bottom panel.

6 - We also found some typos/mistakes left in the text. Find below a non-exhaustive list. Please check the manuscript carefully.

L132: space before the coma.

L145: remove extra space in "FT2 β / FT2 α "

L183/184: please correct the sentence: "A strong interactions were...". It should be either "A strong interaction was..." or "Strong interactions were..."

L188: "Nicotiana. Benthamania"

L231-232: missing words: "...indicate that FT2 β functions as a repressor of the FORMATION OF THE flowering initiation complex..."

L237-238: Please revise the sentence. For example "...; however, we did not detect any physical interaction of FT2 β with FDL2 or 14-3-3s..."

L239: "Thus, we investigated ANOTHER possible mechanism."

L249/L770/L776: remove extra space.

L488 and L812 : should be "were performed at least IN three independent biological..."

Please, also check the legends of supplemental figures carefully.

Responses to Reviewers' Comments

We thank the reviewers for their constructive comments to improve our manuscript. We have revised the manuscript for all concerns. The point by point responses to the reviewers' comments are listed in detail below:

1 - L31/32 : This sentence is confusing, as one may interpret that the change in isoform also acts on flowering in other temperate grasses. Please precise that the mechanism you are referring to is the alternative splicing of FT2 β

Response: Thanks the reviewer for pointing this out. To precise the mechanism as the reviewer suggested, we changed the statement to "Furthermore, we show that the alternative splicing of *FT2* is conserved in important cereal crops such as barley and wheat."

2 - L209-211: The sentence is not clear: the fact that FT2 β does not bind to FD/14-3-3 does not imply that it represses flowering, but that it might act through another mechanism, for example by interfering with the formation of the flowering activating complex. Please revise.

Response: We revised the statement from " FT2 β 's inability to bind to FD and 14-3-3s implied that it represses flowering, perhaps by interfering with flowering activation complex formation." to "FT2 β 's inability to bind to FD and 14-3-3s implied that it may interfere with flowering activation complex formation."

3 - Figure 1 & Figure 6: please indicate the level of significance for * and ** in the legend.

Response: We added the significance level in the revised figure legend.

4 - Note that colors used in Figure 6 (FT2 α vs. FT2 β) are impossible to discriminate for color blind people (or when printed in B&W). Maybe consider using more contrasting colors and/or use different symbols for the gene expression kinetics.

Response: We think that the reviewer may point Figure 5 rather than Figure 6 is difficult for color blind people to discriminate, because there are only distinct

colors for the gene expression kinetics in Figure 5, and we changed it more contrasting in the revised Figure 5 as the reviewer suggested.

5 - Figure S14: the y-axis annotation is lacking on the bottom panel.

Response: We added this in the revised figure S14.

6 - We also found some typos/mistakes left in the text. Find below a non-exhaustive list. Please check the manuscript carefully.

L132: space before the coma.

L145: remove extra space in "FT2 β / FT2 α "

L183/184: please correct the sentence: "A strong interactions were...". It should be either "A strong interaction was..." or "Strong interactions were...".

L188: "Nicotiana. Benthamania"

L231-232: missing words: "...indicate that FT2 β functions as a repressor of the FORMATION OF THE flowering initiation complex..."

L237-238: Please revise the sentence. For example "...; however, we did not detect any physical interaction of FT2 β with FDL2 or 14-3-3s..."

L239: "Thus, we investigated ANOTHER possible mechanism."

L249/L770/L776: remove extra space.

L488 and L812 : should be "were performed at least IN three independent biological..."

Please, also check the legends of supplemental figures carefully.

Response: We checked the whole manuscript and supplemental figure legends carefully and revised all mistakes.